# Lamination, Borders, and Thalamic Projections of the Primary Visual Cortex in Human, Non-Human Primate, and Rodent Brains

**DOI:** 10.3390/brainsci14040372

**Published:** 2024-04-11

**Authors:** Song-Lin Ding

**Affiliations:** Allen Institute for Brain Science, Seattle, WA 98109, USA; songd@alleninstitute.org

**Keywords:** visual cortex, lateral geniculate nucleus, pulvinar, gene markers, development, evolution, lamination, blindsight

## Abstract

The primary visual cortex (V1) is one of the most studied regions of the brain and is characterized by its specialized and laminated layer 4 in human and non-human primates. However, studies aiming to harmonize the definition of the cortical layers and borders of V1 across rodents and primates are very limited. This article attempts to identify and harmonize the molecular markers and connectional patterns that can consistently link corresponding cortical layers of V1 and borders across mammalian species and ages. V1 in primates has at least two additional and unique layers (L3b2 and L3c) and two sublayers of layer 4 (L4a and L4b) compared to rodent V1. In all species examined, layers 4 and 3b of V1 receive strong inputs from the (dorsal) lateral geniculate nucleus, and V1 is mostly surrounded by the secondary visual cortex except for one location where V1 directly abuts area prostriata. The borders of primate V1 can also be clearly identified at mid-gestational ages using gene markers. In rodents, a novel posteromedial extension of V1 is identified, which expresses V1 marker genes and receives strong inputs from the lateral geniculate nucleus. This V1 extension was labeled as the posterior retrosplenial cortex and medial secondary visual cortex in the literature and brain atlases. Layer 6 of the rodent and primate V1 originates corticothalamic projections to the lateral geniculate, lateral dorsal, and reticular thalamic nuclei and the lateroposterior–pulvinar complex with topographic organization. Finally, the direct geniculo-extrastriate (particularly the strong geniculo-prostriata) projections are probably major contributors to blindsight after V1 lesions. Taken together, compared to rodents, primates, and humans, V1 has at least two unique middle layers, while other layers are comparable across species and display conserved molecular markers and similar connections with the visual thalamus with only subtle differences.

## 1. Introduction

Determination of anatomical boundaries and layers of specific cortical regions is critical for correct targeting and sampling as well as reliable comparison and interpretation of experimental results. The primary visual cortex (i.e., first visual cortex, V1), striate cortex, or cortical area 17 is a focus of extensive attention and investigation across species. The layers and boundaries of V1 in adult humans and non-human primates (NHPs) are relatively easy to identify because of the existence of highly specialized middle cortical layers (layer 3 and/or layer 4, depending on different definitions). However, in rodents, specialization of the middle layers is not obvious, or these layers do not even exist, making identification of the layers and borders of V1 more difficult. Additionally, the specialization of the middle layers of V1 is not well established in prenatal humans and NHPs. It is also not clear whether and at which ages the boundaries of V1 can be reliably identified in rodent brains. Since several recent BRAIN Initiative Cell Atlas Network (BICAN) projects (funded by NIH) focus on the comparison of transcriptomic cell types in human, NHP, and rodent brains, this review article, as a starting point of harmonizing brain ontology across these species, focuses on unifying the definition of cortical layers and borders of V1 in these species. Accordingly, other species, such as carnivores, are not included in this article. Molecular markers for cortical layers and major thalamic connections of V1 across species are also compared and discussed in relation to the boundaries of V1. In addition to the related literature, this article also uses some related data derived from the following large open-access datasets after careful evaluation and verification of the data. These include Allen adult human brain dataset (http://www.human.brain-map.org, accessed on 8 January 2024; [1,2]), Allen developing human brain dataset (http://www.brainspan.org, accessed on 8 January 2024; [3,4]), NIH blueprint nonhuman primate brain dataset (http://www.blurprintnhpatlas.org, accessed on 15 January 2024; [5,6]), the Brainmaps.org website (http://www.brainmaps.org, accessed on 15 January 2024; [7,8]), Allen mouse brain dataset (http://www.mouse.brain-map.org, accessed on 18 January 2024; [9]), Allen mouse brain connectivity dataset (http://www.connectivity.brain-map.org, accessed on 18 January 2024; [10]), and the Marmoset Gene Atlas dataset (http://www.brainminds.jp, accessed on 22 January 2024; [11,12]).

## 2. Laminar Organization of V1 in Humans and NHPs 

According to Brodmann, V1 in humans and NHPs possesses a thick and laminated layer 4 (L4; termed inner granular cell layer), which contains many small and densely packed cells in its outer (L4A) and inner (L4C) parts separated by a sparsely packed intermediate part (L4B), which contains large cells. L5 is located immediately below this L4. L5 (termed ganglion cell layer) is thin and occupied by many large ganglion (pyramidal) cells with low packing density. Below L5 is the L6 (termed multiform cell layer), containing cells with a variety of morphology. L6 consists of an outer part with a high packing density of triangular cells (L6A) and an inner part with a low packing density of multiform cells (L6B). L3 (termed pyramidal cell layer) is located immediately above L4. L3 is thick and contains many larger pyramidal neurons. L2 (termed outer granular cell layer) is thin and occupied by small pyramidal cells located between the cell-less L1 (termed molecular layer) and the thick L3 [13]. Brodmann’s L4C can be further subdivided into L4Cα and L4Cβ. These sublayers, together with L4A and L4B, merge into a single L4 at the border with the secondary visual cortex (V2) or cortical area 18 [13]. Note that the capital letters A, B, and C indicate the sublayers of V1 based on Brodmann’s and Hassler’s terminology, while the lower-case letters a, b, and c indicate the sublayers of V1 in the present study. L4A and L4C in macaque monkeys were found to form thin and thick bands of strong cytochrome oxidase (CO) activity, respectively, while L4B shows weak CO activity [14,15]. Similarly, strong Vglut2 immunoreactivity was observed in L4A (thin) and L4C (thick) of the squirrel, vervet, and macaque monkeys, while in the apes and humans, only the latter band in L4C was observed [15,16,17]. Interestingly, in tangential sections cut through the L4A, the CO+ and Vglut2+ L4A band displays a honeycomb-like appearance [14,15,16,17]. 

Although many researchers followed Brodmann’s laminar scheme for V1 of humans and NHPs, some other authors such as Hassler [18], Fitzpatrick et al. [19], Elston and Rosa [20], Takahata et al. [21], and Balaram and Kaas [22] questioned whether Brodmann’s L4A and L4B should really be seen as part of L4, based on comparison of V1 and V2 across NHP species. The evidence that does not support Brodmann’s L4A and L4B as parts of real L4 includes the following: first, the cells in L4B of the primate V1 form strong extrinsic projections to other areas, while real L4 cells do not do this in other areas [23,24,25]. Second, L4A and L4B do not contain as many small granular cells as real L4 does. Third, L4A and L4B do not strongly express the L4-L3b genes such as *Rorb* and *Cux2*. Fourth, L4A and L4B do not receive strong inputs from the lateral geniculate nucleus [LG; also termed dorsal lateral geniculate nucleus (DLG)], as the real L4ab does (for details, see the description below). Finally, it should be mentioned that L4A and L4B appear to be additional layers unique to primate V1, and thus, they also do not belong to typical L3 (L3b). Since Brodmann’s L4A and L4B are located between typical L3b and real L4a, the only reasonable way is to rename them as parts of specialized L3 (i.e., L3b2 and L3c, respectively) to keep the order of cortical layer from L1 to L6, as some authors did previously [19,20,22]. Therefore, the laminar scheme with ten layers (L1, L2, L3A, L3B, L3Bβ, L3C, L4A, L4B, L5A, L5B, L6A, and L6B) was recommended for consistent use in humans and NHPs [22]. 

The results of my additional analysis of publicly available datasets also suggest that the latter scheme is better for comparison between V1 and other cortical areas and across species. Accordingly, I treat Brodmann’s L4Cα and L4Cβ (densely packed with small granular cells) as real L4 (termed L4a and L4b, respectively) and his L4A and L4B as L3b2 and L3c, respectively (see Figure 1 and Figure 2). L3b2 is a simplified term for the L3Bβ of Hassler’s term. L3c and L4ab can be more easily appreciated in parvalbumin (PV)-immunohistochemistry (IHC)-stained sections based on the overall light and dark staining in L3c and L4ab, respectively (Figure 1A,C,D), compared with the Nissl-stained section (Figure 1B). In non-phosphorylated neurofilament (NPNF)-IHC-stained sections, L3c and L4ab show strong and faint staining, respectively (Figure 2A–C). NPNF labeling in L3ab and L5-L6 is moderate, while that in L2 and L3b2 is light in macaque monkeys (Figure 2A,B). In human V1, strong NPNF staining is seen in L3c and L5-L6 but not in L2, L3ab, L3b2, and L4 (Figure 2C). NPNF labeling in the marmoset V1 overall resembles that in human V1 [26]. L4a and L4b are the major recipient zones of the geniculo-V1 projections originating from the lateral geniculate nucleus (LG) [19,27]. L4a and L4b were reported to receive inputs from magnocellular (LGmc) and parvicellular (LGpc) parts of the LG, respectively [19,27]. The stria/band of Gennari roughly corresponds to L3c, which is the relatively lightly stained zone located between L3b2 and L4a in Nissl-stained sections and contains dense myelin labeling in myelin-stained sections. V1 in other NHPs (including simians and prosimians) has a similar laminar scheme [21,22,25] as in human and macaque brains demonstrated in the present study.

## 3. Laminar Organization of V1 in Rodents

In rat V1, the equivalents of primate L3b2 and L3c do not appear to exist [28,29,30]. Like in rats, mouse L4 does not appear to have obvious subdivisions (L4a and L4b) and, in Nissl-stained sections, is more like L4b of the NHPs in terms of the densely packed granular cells (Figure 3A). Furthermore, since the mouse LG (usually termed DLG in the rodent literature; sometimes the primate’s LG is also termed DLG) does not contain an equivalent LGmc, mouse L4 could be the equivalent of L4b in humans and NHPs. Despite this, a single laminar term, L4 rather than L4b, is used in the present study to label the whole inner granular layer to be consistent with the rodent literature (rats [28]; squirrels [31]). Detailed comparative studies at single-cell transcriptomic and connectional levels may be needed to finally resolve this issue. Due to the lack of L3b2 and L3c, rodent V1 has the following layers: L1, L2, L3a, L3b, L4(L4b), L5a, L5b, L6a, and L6b (Figure 3A–F). These layers probably correspond to those with the same laminar labels in humans and NHPs described above. As demonstrated in Nissl-stained sections from the mouse V1 (Figure 3A), L2 immediately underlies the cell sparse L1 and is composed of densely packed small pyramidal cells. L3a and L3b mainly consist of medium-sized pyramidal neurons with less dense packing density compared to L2 and L4. Granular L4 can be easily identified with its small cell sizes and high packing density. L5 has larger cells overall and a lower packing density than underlying L6, which adjoins the white matter. In situ hybridization (ISH)-stained sections for gene *Rorb* (RAR-related orphan receptor beta) show that L5 contains sparsely packed *Rorb*-expressing neurons while overlying L4 strongly expresses *Rorb* (Figure 3B). It is noted that many *Rorb*-expressing cells also exist in L3b but not in L3a, and this could mislead some researchers to treat *Rorb* as a good L4 marker. In NPNF-IHC-stained sections from the mice, labeled neurons are mainly located in L3b and L5 (Figure 3C). This pattern is overall similar to that in NHPs (Figure 2A,B), given that the mouse V1 probably does not have an L3c, which is strongly stained in the monkeys as an additionally labeled band (Figure 2B). Therefore, it is not proper to treat the strong NPNF expression in L3c as evidence for inconsistent lamination between the primates and rodents because no corresponding L3c exists in rodents, including squirrels [30].

The mouse V1 is also characterized by a subset of neurons expressing the genes *Bmp5* (bone morphogenetic protein 5) and *Scnn1a* (sodium channel, non-voltage-gated 1 alpha) in L5 and L4, respectively (Figure 3D,E). Finally, the projections from the LG (i.e., DLG) to V1 of the mice predominantly terminate in L4 and L3b with much less in other layers (Figure 3F). In the mice, L1-4 and L5-6 occupy about half of V1 thickness, respectively (Figure 3), while in the primate V1, L2-4 is about 3–5 times thicker than L5-6 (Figure 1D and Figure 2B). 

## 4. Layer-Specific Molecular Markers for V1 across Species

As described above, V1 in humans and NHPs probably has unique L3b2 and L3c, while other layers appear to correspond to those of rodents. To provide molecular evidence for this claim, several large ISH datasets for human, monkey, marmoset, and mouse brains, as mentioned above, were explored, and some conserved layer-specific gene expression patterns for the corresponding layers across species were observed. The genes with overall conserved expression patterns in V1 of different species include *Rorb*, *Crym* (crystallin, mu), *Syt6* (synaptotagmin IV), *Ntng2* (netrin G2), *Scn4b* (sodium channel, type IV, beta), *Tle4* (transducing-like enhancer of split 4) and *Pdyn* (prodynorphin). 

As shown in Figure 4, the left column (Figure 4A,G,M,S) illustrates the layers of V1 in Nissl-stained sections, while other columns show specific gene expression patterns revealed with ISH across species. Specifically, *Rorb* is strongly expressed in L4 and L3b across the species examined, with additional strong expression in L3a in humans and macaque monkeys (Figure 4B,H,N,T). Weaker *Rorb* expression is also seen in L5 of the adult human, macaque, and mouse brains but not yet in L5 of the newborn (P0) marmosets (Figure 4N; *Rorb* expression data in adult marmosets are not yet available). Expression of *Rorb* is not detected in L6 of all species examined, while *Rorb* expression in the unique L3b2 and L3c of V1 in humans and NHPs is the weakest compared to other layers (Figure 4B,H,N). Similarly, the upper layer marker *Cux2 (cut-like homeobox 2)* is strongly expressed in L2, L3a, L3b, and L4 of the human, marmoset, and mouse V1 with no expression in L5 and L6. Again, the unique L3b2 and L3c have faint *Cux2* expression. Therefore, L3b2 and L3c in humans and NHPs do not express L4 and L3b marker genes such as *Rorb* and *Cux2*, supporting the renaming from L4A and L4B to L3b2 and L3c, respectively, to harmonize laminar scheme across species [19,21,22].

As for other layers, *Crym* expression is observed in L5 and L6 of V1 across species (Figure 4C,I,O,U). In the human V1, *Crym* is also strongly expressed in L3 and L6 with faint expression in L2, L3b2, L3c, and L4ab (Figure 4C). In macaque V1, *Crym* is expressed in L5 and L6a, and faint expression is present in L3b2, L4ab, and L6b (Figure 4I). In newborn marmosets, *Crym* expression is relatively stronger in L5b and L6a of V1 compared to faint expression in other layers (Figure 4O). In adult mice, strong *Crym* expression is seen in L5 and L6, with no expression in L2-4 (Figure 4U). *Syt6* (Figure 4D,J,P,V) and *Tle4* (e.g., Figure 4R) are strongly and exclusively expressed in L6 across species. *Ntng2* is expressed in L5 and L6a of the human V1 (Figure 4E), in L5 (weaker) and L6a (stronger) of the macaque V1 (Figure 4K), in L6 of the marmoset V1 (Figure 4Q), and in L6 (strong) and L2-4 (weaker) of the mouse V1 (Figure 4W). *Scn4b* is expressed in L3b2-L3c and L5 of the human V1 (Figure 4F), in L3ab, L3b2-L3c, and L5b of the macaque V1 (Figure 4L), and in L5 of the mouse V1 (Figure 4X). *Scn4b* is not expressed in newborn marmoset V1. 

Finally, *Pdyn* is consistently expressed in L5a of V1 and V2 in all species examined (Figure 5A–F). However, in other layers, *Pdyn* is differentially expressed in V1 versus V2 across species. For instance, *Pdyn* is not expressed in L3c, L4a, and L4b of the human V1, but it is expressed in all layers of V2 (Figure 4A,B). In the macaque, in addition to the expression in L5a of V1 and V2, *Pdyn* is also expressed in L3b2 of V1 but not of V2 (Figure 5C,D). In the marmoset, *Pdyn* is expressed in both L5a and L6 of V1 and V2, as well as in L2 and L3ab of V1 (Figure 5E,F); faint *Pdyn* expression is seen in L3b2 of V1 and in L2 and L3a of V2 (Figure 5F). In the mouse V1, *Pdyn* is also expressed in L5a with sparse expression in L3 and L4 (Inset in Figure 5E).

Taken together, some molecular markers for the layers of V1 (e.g., *Rorb*, *Cux2*, *Crym*, *Syt6*, *Ntng2*, *Scn4b*, and *Pdyn*) show overall conserved expression patterns across species with only slight differences. 

## 5. V1 Borders with V2 and Prostriata in Adult Humans and NHPs

In adult humans and NHPs, V1 always adjoins V2 (Brodmann area 18) along its anterior–posterior (A–P) extent except at the most anteromedial location where it abuts area prostriata (Pro) [2,32,33,34]. V1 borders with V2 and prostriata are easily identifiable in humans, and most (if not all) NHPs using the neurochemical markers such as Vglut2, PV, NPNF, Occ1, 5HT1B, and 5HT2A ([2,15,16,17,21,22]; also see Figure 1 and Figure 2A,B) as well as in Nissl-stained sections ([33]; also see Figure 5A,C,E). This is based on the characteristic cytoarchitecture of V1, such as very thick granular L4 and the existence of L3b2 and L3c, as described above. In addition, many molecular markers selectively (strongly or weakly) expressed in V1 can make V1 borders stand out from the adjoining V2 and prostriata in adult humans [35,36], macaques [5,37,38,39], and marmosets [34,40].

## 6. Boundary Determination of V1 in Prenatal Humans

In contrast to adult human V1, V1 borders in prenatal human brains are difficult to identify based only on cytoarchitecture before the unique L3c and the thick L4ab become differentiable. A recent study shows that V1 borders can be clearly delineated on the base of some region-specific molecular markers at certain developmental stages. For example, in the human brains at postconceptional week (PCW) 21, genes *Enc1 (ectodermal-neural cortex 1), Gap43* (growth-associated protein 43), and *Lmo4* (LIM domain only 4) show much less expression in V1 than in adjoining V2 while gene *Npy* (neuropeptide Y) has much higher expression in V1 than in V2 [4]. An example of *Enc1* expression along the A–P extent of V1 and adjoining regions at PCW 21 is shown in the prenatal human brain atlases ([4]; see pages 445–448 for *Enc1* expression). *Npy* expression in V1 and V2 at PCW 21 is displayed in Figure 6A of this article. It is noted that *Npy* expression at PCW 21 is concentrated in the deep portion of the outer cortical plate (CPo; Figure 6B,C). It is worth mentioning that the V1 boundaries revealed with these markers are consistent and almost clear-cut at this early developmental stage (PCW21). From embryonic week 29 onward, the human V1 can be clearly identified in Nissl-stained sections based on the appreciable thick L4 [41,42].

## 7. Boundary Determination of V1 in Prenatal Macaque Monkeys

Like in prenatal humans, L4ab and L3c of the macaque V1 are not well developed during early prenatal development. To explore if V1 in prenatal macaque monkeys can be clearly identified, *Enc1* expression patterns at embryonic day 60 (E60), E70, E80, and E90 of the NIH Blueprint NHP atlas [6] were analyzed. V1 boundaries along the A–P extent can be clearly delineated at E80 and E90 (Figure 7) but not at E60 and E70. At E90, *Enc1* expression in the superficial layers (L1-4) of the monkey V1 is much weaker than those of adjoining regions (mostly V2), making V1 stand out along full A–P extent (Figure 7A–L). The staining intensity of *Enc1* expression in the deep layers (L5-6) of V1 is comparable with adjoining regions such as V2 (Figure 7). It should be mentioned that by E120, the border of the monkey V1 with adjoining V2 can be clearly identified on Nissl-stained sections. Differential expression of the Eph/ephrin genes was also reported between V1 and V2 of the prenatal macaque brains [43]. 

## 8. Boundary Determination of V1 in Newborn Marmosets

Gene expression data (raw data) are available for newborn but not prenatal marmoset monkeys [11,12]. In contrast to prenatal humans and macaque monkeys, strong *Enc1* expression in the newborn marmosets is observed in L2-3 of V1 and adjoining areas. In addition, no and much weaker *Enc1* expression is detected in L4 and L5-6, respectively (Figure 8A–F). V1 in the newborn marmosets is characterized by a thick L4 with no *Enc1* expression and a thin L3c with strong *Enc1* expression (Figure 8C–F). The neighboring V2, posterior cingulate cortical area 23 (A23), and prostriata do not have L3c and display a narrower and faintly labeled L4 (perhaps containing some *Enc1* expression), while the retrosplenial cortical areas 29 and 30 (A29 and A30, respectively) show no clear L4, as indicated by the lack of a “gap” zone between L2-3 and L5-6 (Figure 8A,B). In addition, the superficial part of V2 has a stronger *Enc1* expression than that of V1 (e.g., Figure 8E,F). This pattern is overall similar to that in the macaque V1 at E90 (Figure 7), although *Enc1* expression in the superficial layers of V1 is stronger in marmosets than in the macaques. Therefore, the borders of V1 along the A–P extent can be clearly identified at newborn marmosets (Figure 8C–F). 

V1 boundaries of the newborn marmosets can be further confirmed with other molecular markers such as *Rorb* and *Npy*. As shown in Figure 9A–F, *Rorb* is strongly expressed in L4 of V1 and V2 and moderately in L3b-L3b2 of V1, which do not exist in V2 (Figure 9C–F). The unique L3c of V1 is located between the L3b2 and L4ab and displays only faint *Rorb* expression. In addition, Figure 10 shows the strong *Npy* expression in L3c of V1, which does not exist in V2, thus making V1 stand out from V2 (Figure 10A–F). Additionally, a band of *Npy* expression is also seen in the L2 of the prostriata and the most anterior part of V1 (Figure 10B–E). It is noted that the subplate zone (SP) and developing white matter (or intermediate zone, IZ) of V1 and nearby cortical regions contain sparsely distributed but strongly labeled *Npy*-expressing cells (Figure 10A–F). Finally, many other genes with differential expression between V1 and V2 of the newborn marmosets were reported including *Btbd3, Ctgf, Tbr1, Rora, Nr1a1, Foxp2, Epha6, Epha7, Epha5, Cdh8, Sema6a, Nr4a2*, and *Er81* [44].

## 9. Boundary Determination of V1 in Developing Mice 

Compared to humans and NHPs, V1 boundaries in both developing and adult mice are not easily identifiable in Nissl preparations due to the lack of L3b2 and L3c (see Figure 3). However, expression patterns of some genes in V1 are different from those in adjoining regions. For example, although the *Enc1* expression pattern in the mouse V1 is not clearly distinguishable from that in the adjoining cortices at the prenatal ages, such as E18.5 (Figure 11A), *Npy* expression in V1 is different from nearby cortices. Specifically, *Npy* is mostly expressed in the developing L6 of V1 with no or less expression in the upper layers (Figure 11B). At postnatal day 4 (P4), *Enc1* expression in V1 is different from posteriorly adjoined cortical regions (i.e., postrhinal cortex (PoR) and parasubiculum (PaS); see Figure 11C). The *Npy* expression pattern at P4 (Figure 11D) is similar to that at E18.5. Specifically, at P4, *Npy* is mostly expressed in L6 of V1, while in the primary somatosensory cortex (S1), *Npy* is expressed in both L6 and the upper layers. In the anterior V2 and/or posterior parietal cortex (PPC), *Npy* displays an overall weaker expression. At the posterior levels, almost no *Npy* expression is seen in the PoR–PaS (Figure 11D). Therefore, this *Npy* expression pattern makes V1 identifiable from adjoining regions. 

## 10. Boundary Determination of V1 in Adult Mice 

There are some genes that are selectively expressed in specific layers of the mouse V1. For instance, *Bmp5* (Figure 3D) and *Scnn1a* (Figure 3E) are expressed in L5 and L4 of V1, respectively, with no or much less expression in V2 from P14 to adult mice (http://www.brain-map.org), making boundary identification of V1 much easier. In addition, *Scnn1a* is also mostly expressed in L4 of the S1 and primary auditory cortex (A1) (http://www.brain-map.org). Like *Scnn1a* (Figure 3E), *Scnn1a-Cre* (*tdTomato)* expression is mostly expressed in L4 of V1, S1, and A1. Thus, precise V1 (as well as S1 and A1) boundaries can be reliably and consistently revealed with this marker along the A–P levels of the cortex (see Figure 8A–H). Specifically, at the anterior and middle levels, V1 is located on the dorsal aspect of the cortex and is bordered by V2m medially and V2l laterally (Figure 12A–E). However, at the posterior levels where V2m disappears, the medial V1 extends ventrally into the medial aspect of the cortex and directly adjoins the prostriata (Pro) while the lateral V1 is still bordered by V2l (Figure 12F–H). Moreover, the expression of the marker gene for L5 of V1 (*Bmp5*; see Figure 3D) further confirmed the borders and topography of the mouse V1 **(**http://www.brain-map.org). It is important to point out that *Bmp5* is only expressed in L5 of V1 but not of the S1 and A1, and thus, *Bmp5* appears to be the only V1-L5-specific gene in mice reported so far. Detailed topography of the mouse V1 with the dorsal and ventral subdivisions of the prostriata has been demonstrated recently [45]. Briefly, the mouse V1 is mostly surrounded by V2 (V2m and V2l) except at the most posteromedial levels, where it abuts the prostriata. The location and extent of the posteromedial extension, as well as the middle and medial V1, are also delineated in sequential sagittal sections based on the expression pattern of the same marker *Scnn1a-Cre* (*tdTomato)* to serve as the reference plates for lateral-to-medial sagittal sections (Figure 13A–I). In addition, *Scnn1a* is also expressed in the visual thalamus, LG (DLG; Figure 13D). Unfortunately, *Bmp5* and *Scnn1a* expression data for humans and NHPs are not available for comparison. However, fortunately, V1 borders for humans and NHPs can be reliably identified in Nissl-stained sections without molecular markers. It should also be mentioned that the prostriata of the humans and NHPs are located anterior to the anterior mediodorsal V1 [2,32,33,34]. Compared to the location of the border between the rodent V1 and the protriata ([45,46]; also see Figure 12), the shift of the border from the posteromedial (rodents) to anteromedial (primates) locations likely reflects the rotation of V1 from overall dorsal (rodents) to overall ventromedial (primates) locations.

Importantly, the posteromedial extension (between the arrows in Figure 12F,G) of the medial V1 in rodents obviously occupies the region previously treated as the retrosplenial cortex (RSA/A30 and RSG/A29) and/or V2MM/RSPagl [30,47,48,49]. The existence of the posteromedial V1 extension in these regions suggests that the mislabeled region in the rodent atlases likely needs to be revised. In addition, the region V2MM/RSPagl has recently been found to be the equivalent of the primate posterior cingulate area 23 (A23; see [50]). In summary, the rodent V1 was typically defined on the dorsal aspect of the visual cortex in the literature, negating the existence of the posteromedial V1 extension, which strongly expresses V1 marker genes such as *Scnn1a* and *Bmp5* (see Figure 3 and Figure 12) and connects with the prostriata [46,51] and LG (i.e., DLG; see below).

## 11. Thalamocortical Projections of V1 in Human and NHP Brains

For human brains, methods for direct tracing of neural connections are not yet available. However, in high-resolution histological and MRI images of the same human brain, a major part of the LG-V1 projection pathways (i.e., optic radiations or the external part of the sagittal stratum) can be clearly identified and followed from the region near the LG to the white matter region underlying V1 (see Figure 19 of [2]). Histologically, many of the axonal profiles in the optic radiations are parvalbumin-positive (e.g., Figure 7D of [2]). Different DTI tractography methods were also used to trace the optic radiation in vivo in human brains with cadaveric anatomy of the optic radiations as a golden standard reference [52,53,54]. It may be more helpful to use histological markers such as NPNF (negative marker) or parvalbumin (positive marker) as an anatomical reference of the optic radiations since both markers enable clear and accurate identification of the optic radiations along the A–P axis (see sequential plates in [2]). Optic radiation can also be identified with its parvalbumin-positive axons in marmoset monkeys [55].

As for the NHP brains, the LG was initially reported to exclusively project to V1 [56,57,58,59,60,61]. Later, many reports showed the existence of the LG projections to the extrastriate or prestriate visual cortices [62,63,64,65]. The projections from the LG to the middle temporal area (MT) in NHPs were also reported [66,67]. Generally, the LG-V1 projections (optic radiations) are strong and organized in a topographical or point-to-point fashion [68,69]. In contrast, LG projections to the extrastriate cortices are relatively weak, not organized in a point-to-point fashion, and mainly originated from the interlaminar and S layers of the LG [64,65]. In NHPs, as mentioned above, LGmc and LGpc projects to L4a and L4b of V1, respectively. In addition, LGpc also projects to L3b2 of V1 [19,25,27,70]. Finally, the intercalated/koniocellular layers of the LG were reported to project to L3b (in patches/blobs) and L1 of V1 [19,25,57,71,72,73,74]. The L3b blobs are positive for cytochrome oxidase (CO; [14,15]). L3c (i.e., Brodmann’s L4B) does not appear to receive inputs from LG. Instead, L3c appears to receive inputs mainly from L4a of V1 [75] and the CO-reactive stripe of V2 [76]. 

Finally, compared to simians, the basic organization of the projections from LGmc, LGpc, and LGko to V1 in prosimians (e.g., Galago) is similar [77]. However, some differences were also reported. For example, the LG-V1 projections in Galagos are compressed in comparison to that in monkeys, and L3c in Galagos is incipient [22,25]. In addition, L3b in prosimians lacks a projection from LGpc [25]. 

## 12. Corticothalamic Projections of V1 in Human and NHP Brains

V1 of the NHPs projects to the LG, pregeniculate nucleus (PG), laterodorsal thalamic nucleus (LD), lateroposterior thalamic nucleus (LP), pulvinar (Pul), superior colliculus (SC) and pontine nucleus (PN) [78,79]. Corticothalamic projections from the primate V1 to the LG were reported to originate from both L6a and L6b [80,81] and terminate in all layers of the LG [78,80,82]. More specifically, L6a of V1 projects to LGpc, while L6b mainly projects to LGmc [80,81]. However, some neurons in L6 of V1 (including the large Meynert cells) also project to the SC and MT [83,84]. The projections from V1 to other subcortical regions mainly originate from L5 neurons and mostly innervate the PG, LD, LP, Pul, SC, and PN [79,80]. V1 projections to the LG, LD, LP–Pul, and SC display topographic organization [79].

## 13. Thalamocortical Projections of the Rodent V1

Like in the primates, rodent V1 receives strong inputs directly from the equivalent of the primate LG (i.e., DLG) [28,46,85,86,87,88]. The axonal terminals of these projections strongly target L4 and deep L3 and weakly terminate in L1 and L6 of V1 [28,46,89]. Weaker projections were also reported in the medial and lateral V2 of the rats [89]. However, a recent study in rats and mice demonstrated that the most anterior part of the LG (DLG) also sends moderate projections to the dorsal subdivision of the prostriata [46], which directly adjoins the posteromedial V1 extension ([45]; also see Figure 12 and Figure 13). The LG-prostriata projections appear to exist in human brains [90] and have recently been proposed to contribute to the “blindsight” observed in patients with V1 lesions [46].

To further explore whether the LG (DLG) projections target V1 or both V1 and V2 of the mice, several Cre-line mice with the anterograde tracer (rAAV) restricted in the LG (DLG), which are available from the Allen mouse brain connectivity dataset, were analyzed. Figure 14 shows two rAAV injections restricted in the LG (DLG) of the *Slc17a6*-Cre (Figure 14A–D) and *Scnn1a*-Cre (Figure 14F–H) mice. Both injections are similarly restricted in the ventrolateral part of the anterior and intermediate LG (DLG) with no involvement in the dorsomedial part (indicated by the arrows in Figure 14) and the posterior part (more posterior to level D/H of Figure 14) of the LG (DLG). *Scnn1a*-Cre (Figure 13D) and *Slc17a6*-Cre (Figure 14E) are strongly expressed in the DG (DLG) but not in the PG (VLG). As expected, the labeled axon terminal fields from these two injections are very similar (Figure 15). Overall, the labeled terminal fields in both cases concentrated in the medial V1, including the posteromedial V1 extension along the A–P extent (see the insets in Figure 15A and Figure 15I, respectively). The strongly labeled LG terminal field in the posteromedial V1 extension is consistent with the medial V1 border revealed with *Scnn1a* (see Figure 12). Figure 15A–H (from the *Slc17a6*-Cre mouse) demonstrates the terminal distribution in the posterior V1 (indicated in the inset of Figure 15A) in sequential sections with an interval of 100 µm. Figure 15I–L (from the *Scnn1a*-Cre mouse) shows the sections corresponding to Figure 14E–H, respectively. It is obvious that the labeled axon terminal bands are densely distributed in L4 and deep L3 (L3b), with much weaker labeling in L6 and L1 of V1 and in L4 of the lateral V2. These Cre-dependent tracing results indicate that the LG (DLG) projects very strongly to L4 and L3b of V1 (including the posteromedial V1 extension) and very weakly to V2. This finding further supports the existence of the posteromedial V1 extension revealed with strong *Scnn1a* expression in L4 of V1 (see Figure 12 and Figure 13). Therefore, the posteromedial extension of the visual cortex indeed belongs to V1 rather than to the retrosplenial cortex and medial V2.

## 14. Corticothalamic Projections of the Rodent V1

Rodent V1 projects to many subcortical regions, and these projections originate mainly from L5 and L6. L5 of V1 projects to the PG (VLG), LD, LP, SC, PN, and pretectal nucleus (PTN) [43,89,90]. In contrast, V1-thalamic projections appear to exclusively originate from L6 and mainly terminate in the LG (DLG) [91,92]. This is consistent with the organization principle that rodent L6 in different cortical areas predominantly targets their specific thalamic nuclei. For instance, L6 of the A1 projects mostly to the medial geniculate nucleus (MG; [93,94]). L6 of the S1 projects mainly to ventroposterior nuclei [95,96]. Interestingly, the polymorphic layer of the subiculum and prosubiculum (Spo and PSpo, respectively), which probably is the equivalent of neocortical L6 in terms of transcriptomic similarity [97], also originate the projections to the thalamic nuclei (mainly the anteroventral and anteromedial nuclei; see [97]). An additional feature of the corticothalamic projections from most of the neocortical regions is that they also send projections to the reticular thalamic nucleus (Rt) and association thalamic nuclei such as the LP–Pul and LD in addition to their specific thalamic nuclei [91,92]. V1-thalamic projections to the LG, LD, and LP–Pul of the *Ntsr1-*Cre mice are demonstrated in Figure 16 and Figure 17. Like *Syt6* and *Tle4*, *Ntsr1* (neurotensin receptor 1) is exclusively expressed in L6 of the mouse neocortex, including V1 (http://mouse.brain-map.org).

Figure 16 displays the distribution of the labeled axon terminals in the thalamus following an rAAV injection into the caudal-intermediate part of V1 (V1-ci in Figure 16A,B) of a *Ntsr1*-Cre mouse. Although moderate terminal labeling is seen in the lateral part of the LD (Figure 16C), strong terminal staining is concentrated in the rostral-lateral LG (DLG), lateral-dorsal LP–Pul adjoining the LG (Figure 16D–G), and the Rt (Figure 16E). The caudal LG (DLG) and caudal LP–Pul contain almost no terminal labeling (Figure 16H and more caudal levels). 

To further examine possible topographic corticothalamic projections, the terminal distribution in the thalamus of three *Ntsr1*-Cre mice was examined. The three cases have injections located in caudal-medial V1 (V1-cm; Figure 17A–E), caudal-lateral V1 (V1-cl; Figure 17F–J) and rostral-medial V1 (V1-rm; Figure 17K–O). Although the labeled terminals are observed in the LD, LG/DLG, LP–Pul, and Rt in all cases, the precise locations within these structures vary among cases. With a V1-cm injection (Figure 17A), the labeled terminals are mainly seen in the dorsal part of the LD (Figure 17B), ventral part of the LG/DLG (Figure 16C,E), medial part of the Rt (Figure 17B,C) and lateral part of the LP–Pul (Figure 17C,D). In contrast, following a V1-cl injection (Figure 17F), the terminal labeling mainly exists in the lateral LD (Figure 17G), lateral LP–Pul (Figure 17H,I), dorsal-lateral LG/DLG (Figure 17I,J) and medial Rt (Figure 17H), and this pattern is similar to that in the case with V1-ci injection (Figure 16). Finally, with a V1-rm injection (Figure 17K), the labeled terminals are mainly found in the ventral part of the LD (Figure 17L), ventral and medial part of the LP–Pul (Figure 17M,N), lateral part of the Rt (Figure 17M), and ventral-medial part of the posterior LG/DLG (Figure 17N,O). In summary, the corticothalamic projections of V1 display topographic organization. Overall, medial-lateral parts of V1 project to lateral-medial parts of the LP–Pul and ventral-dorsal parts of the LG (DLG), respectively. Rostral-caudal parts of V1 project to ventral-dorsal parts of the LP–Pul, ventromedial-dorsolateral parts of the LG (DLG), and lateral-medial Rt, respectively.

Taken together, the main targets of the corticothalamic projections from rodent V1 are similar to those from primate V1 (see Section 12).

## 15. Possible Neural Circuits Underlying Blindsight

Damage to V1 causes blindness by severing the main pathway from the LG (DLG) to the cortex. However, some visual abilities remain without visual awareness (i.e., residual vision or unconscious vision), which is termed blindsight. Blindsight is hypothesized to be mediated by several pathways that bypass V1. These pathways could include those from the pulvinar and LG (DLG) to the extrastriate cortices [98,99]. However, The LG (DLG) has been considered the most crucial component that supports blindsight via its projections to the extrastriate cortices [99,100,101]. For example, Schmid et al. [100] showed that before LG inactivation in the macaque monkeys, high-contrast stimuli presented to the lesion-affected visual field produced significant V1-independent fMRI activation in the extrastriate cortical areas such as V2, V3, V4, V5/MT, and the animals correctly located the stimuli in a detection task. However, following reversible inactivation of the LG in the V1-lesioned hemisphere, fMRI responses and behavioral detection were abolished. Another important finding is the direct and strong projections from the LG (DLG) to the prostriata in rodent [46] and human brains [90], the latter pending confirmation with higher resolution methods. In normal monkeys, relatively weaker projections were reported from the LG to the extrastriate cortices [62,63,64,65,66,67]. Following V1 lesions, many LG (DLG) neurons were degenerated, but some still survived and were functional [102,103]. In addition, V1 lesions could lead to neurochemical and structural remodeling of the LG-extrastriate pathway, such as the emergence of a pathway that brings information to MT from cell populations that would normally project to V1 and changes in the ratio of GABAergic neurons in the LG (DLG) [104,105]. There was a report in monkeys that following V1 lesions, the surviving LG neurons were mostly those that directly project to extrastriate cortices [106]. Therefore, direct visual thalamo-extrastriate projections [66,67] may contribute to the neural circuits underlying the blindsight after V1 lesions. In addition to the LG (DLG) pathways, the projections from the superior colliculus to the pulvinar may also participate in visuomotor processing, while lateral intraparietal regions are critical in the saccade control in blindsight [107]. Interestingly, a recent study has indicated that despite robust subcortical responses to visual stimulation, little evidence was found for strengthened subcortical input to V5/MT after V1 lesion [108]. Finally, while LG-prostriata projections were not investigated in monkeys, a study in humans indicated the existence of these projections [90]. This finding suggests that LG-prostriata projections may be another contributor to blindsight in humans and monkeys. 

In rodents, LG (DLG) normally sends direct and strong projections to the prostriata, bypassing V1 [46], and the prostriata converges multimodal and bilateral sensory information from many sources [46,109,110]. However, it is not known if rodents have blindsight after the V1 lesion and if rodents have stronger thalamo-extrastriate projections. Nevertheless, it is reasonable to speculate that, after V1 lesions, the LG-prostriata projections, as well as other subcortical-extrastriate projections, would be enhanced to compensate for the loss of many LG-V1 projections. This possible plasticity could be investigated in future studies using rodents as a model. 

## 16. Laminar Development and Vision-Related Functional Maturation and Disorders

As shown in this article, the specialized L4 is detectable from PCW 21 onward in human V1 (Figure 6), from E70 onward in macaque V1 (Figure 7), and at around birth in marmoset (Figure 8, Figure 9 and Figure 10) and mouse V1 (Figure 11) based on specific gene expression. In human brains, adult-like lamination of V1 is identifiable in Nissl-stained sections from prenatal weeks 29–30 onward [41,42]. In general, anatomic changes continue throughout the postnatal developmental period across species. In the monkey’s visual cortex, for example, the laminar distribution of feedback connections changes in the first two months of life [111,112]. Similarly, vision-related functions also mature throughout postnatal developmental stages with basic receptive field (RF) properties, and visual functions mature earlier than complex ones [112]. In the human visual cortex, the fundamental RF architecture becomes adult-like by age 5, and visuo-spatial functions continue to develop afterward. This finding suggests that, despite the early maturation of the RF structure, functional interactions within and across RFs may mature slowly [113].

Laminar aberrancies in development have been associated with an animal model for autism [114], and with disruption of genes associated with developmental dyslexia [115]; both autism and dyslexia display obvious dysfunctions in visual skills. For example, embryonic disruption of the candidate dyslexia susceptibility gene homolog Kiaa0319-like results in neuronal migration disorders [116]. These peculiarities at the laminar level are good candidates to constitute the anatomical counterpart of the functional age-related aberrancies that have been demonstrated in the dyslexic brain [117]. On this basis, it could be proposed that the laminar characteristics described in the present study may be considered part of the anatomical foundations of subsequent functional development, as suggested by the case of dyslexia.

In humans and NHPs, the magnocellular (M) pathway is the major stream of inputs from the retina to LGmc, to V1, and then to the dorsal extrastriate and parietal regions. This M pathway mediates the ability to rapidly identify letters and their order because they control visual guidance of attention and eye fixations. Abnormal development of this pathway could cause dyslexia. Evidence for M cell impairment has been reported at all levels of the visual system [118,119]. In addition, treatments that facilitate M function, such as viewing text through yellow or blue filters, can greatly increase reading progress in children with visual reading problems [119]. Since cell loss in LGmc has been reported in patients with dyslexia [120], and LGmc mainly projects to L4a (see Section 11), it would be interesting to explore in the future whether the thickness of L4 decreases as the disorder progresses.

## 17. Conclusions

Based on the comparative analysis above, the borders of V1 in adult humans and NHPs can be easily identified, while those in prenatal primates and rodents can be more accurately delineated with the help of additional molecular markers. Compared to rodents, V1 in humans and NHPs has at least two unique layers (L3b2 and L3c). If one ignores the unique layers in primate V1, other layers in V1 are overall comparable across species in terms of anatomic features, conserved molecular markers, and reciprocal connections with the visual thalamus (Figure 18). Based on harmonized criteria for V1, a previously mislabeled part of the rodent V1 is uncovered, and this is the posterior ventromedial extension of V1, which was treated as the retrosplenial cortex and/or medial secondary visual cortex in rodent literature and brain atlases. 

## Figures and Tables

**Figure 1 brainsci-14-00372-f001:**
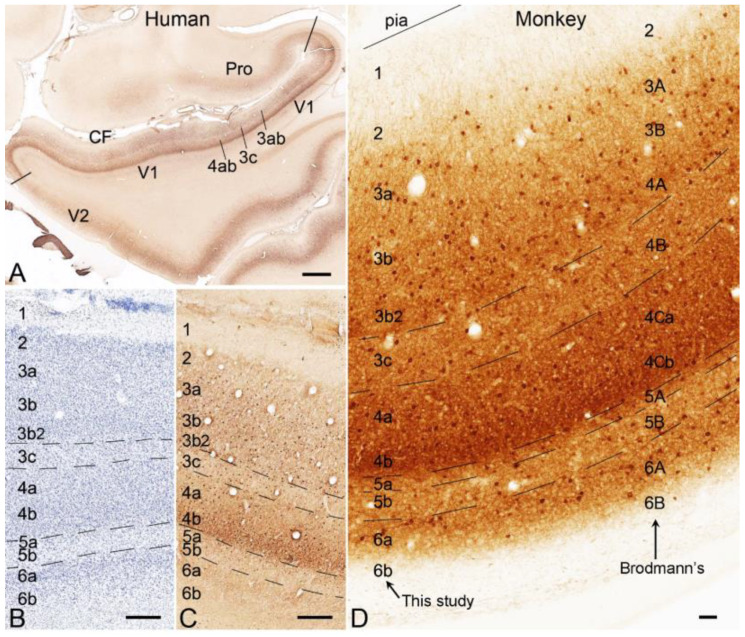
Parvalbumin (PV) immunoreactivity in V1 of adult human and monkey brains. Cortical layers are indicated by Arabic numbers; CF stands for the calcarine fissure. (**A**) A low magnification view of PV staining patterns in the anterior part of V1, V2, and area prostriata (Pro) in a human brain. (**B**,**C**) higher magnification views of the cortical layers in closely adjacent Nissl- (**B**) and PV-stained (**C**) sections of the human V1. (**D**) A high magnification view of the cortical layers in a PV-stained section of V1 from a macaque monkey. The current and Brodmann’s terminologies for the cortical layers of V1 are shown on the left and right sides of the image, respectively. The raw images for panels (**A**–**C**) are derived from the Allen Human Brain Reference Atlas [2], while the raw image for (**D**) is from the website (brainmaps.org). Bars: 1550 µm in (**A**); 390 µm in (**B**,**C**); 45 µm in (**D**).

**Figure 2 brainsci-14-00372-f002:**
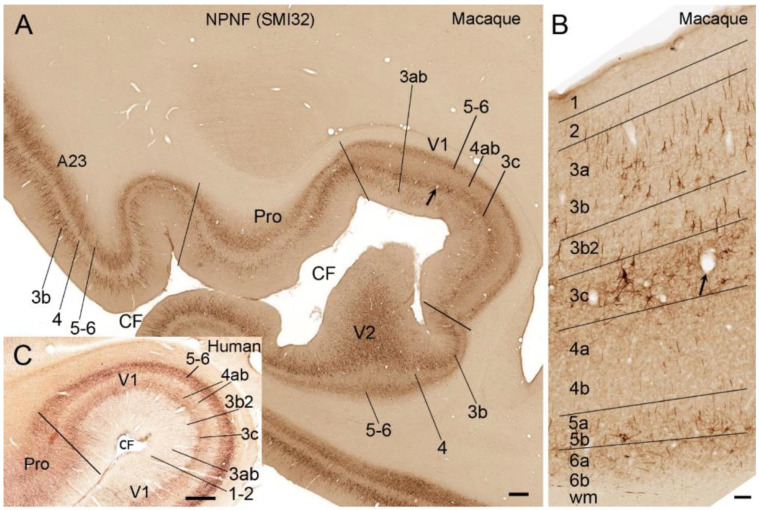
Immunoreactivity for non-phosphorated neurofilament (NPNF) in V1 of the adult macaque and human brains. (**A**) A low magnification view of NPNF staining patterns (labeled with SMI32 antibody) in the anterior part of V1, V2, and prostriata of a macaque monkey brain. Posterior cingulate area 23 (A23) is also shown. (**B**) A high magnification view of the cortical layers in an NPNF-stained section of V1 from the same section as in (**A**). The arrows in (**A**,**B**) point to the same location. Note that stronger NPNF staining is mainly seen in layer 3ab (L3ab), L3c, and L5-6. (**C**) A low magnification view of NPNF staining patterns in the anterior part of V1, V2, and prostriata of the same adult human brain, as shown in Figure 1A. The raw images for panels (**A**,**B**) are derived from the website (www.brainmaps.org), while the raw image for (**C**) is from the Allen Human Brain Reference Atlas [2]. Bars: 370 µm in (**A**); 45 µm in (**B**); 155 0µm in (**C**).

**Figure 3 brainsci-14-00372-f003:**
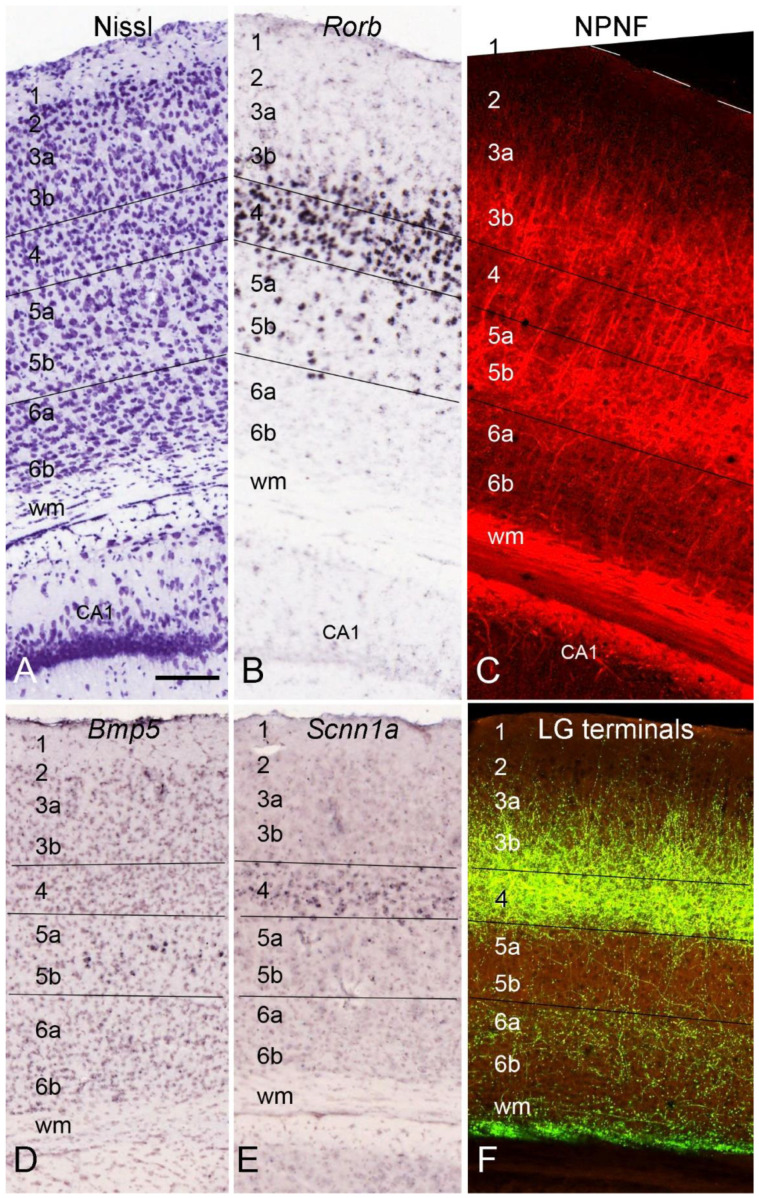
Cytoarchitecture and layers of the mouse V1. (**A**,**B**) Two adjacent sections stained for Nissl substance (**A**) and the gene *Rorb* (**B**) showing the cytoarchitecture (**A**) and *Rorb* expression pattern (**B**) of V1. (**C**) NPNF staining pattern of V1 revealed with SMI32 antibody. NPNF-positive cell bodies are predominantly observed in L3 and L5. (**D**,**E**) Expression of the genes *Bmp5* (**D**) and *Scnn1a (***E**) as molecular markers for L5 and L4, respectively. (**F**) Axon terminal fields of the dorsal lateral geniculate (LG) projections in L4 and L3b are strong with less labeing in L6 and much less in L5 and L3a. The raw images for all panels are derived from the Allen Institute website (www.brain-map.org). Bar: 110 µm in (**A**) for all panels.

**Figure 4 brainsci-14-00372-f004:**
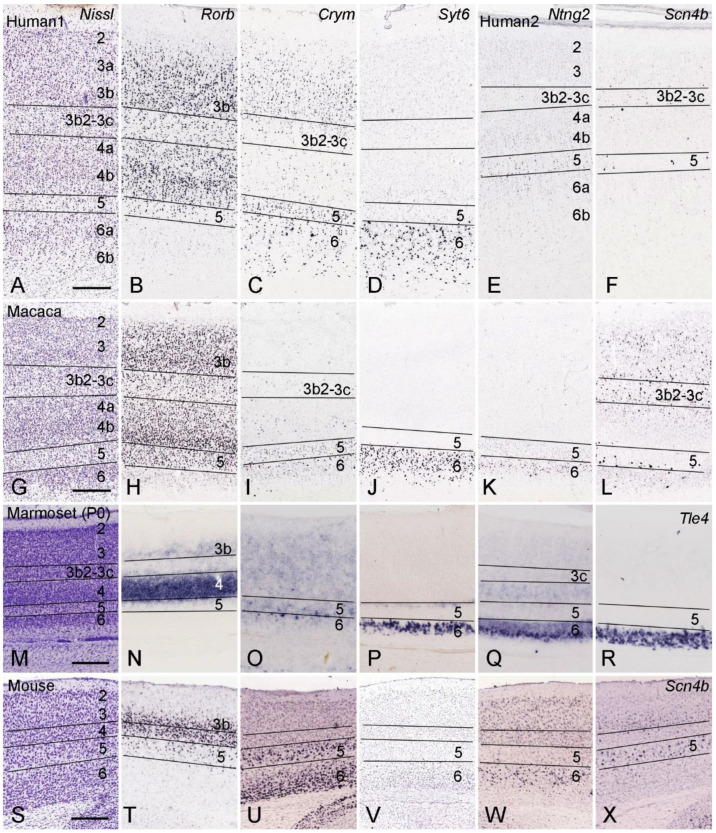
Comparison of some gene markers for cortical layers of V1 across species. Adult human (**A**–**F**), adult macaque monkey (**G**–**L**), newborn marmoset (**M**–**R**), and adult mouse (**S**–**X**) data are shown. The six columns show Nissl, *Rorb, Crym, Syt6, Ntng2,* and *Scn4b* stained sections, respectively, except for panel R, which is replaced with *Tle4* because *Scn4b* is negative at newborn (P0). Overall, conserved strong expression across species is seen for genes *Rorb* (L4 and L3b), *Crym* (L5-6), *Syt6* (L6), *Ntng2* (L6), *Scn4b* (L5), and *Tle4* (L6). Differential gene expression in certain layers is also observed, such as additional *Rorb* expression in L3a of the human and macaque brains, additional *Crym* in L3ab of the human brain, additional *Ntng2* in L5 of human and macaque brains, and additional *Scn4b* in L3ab and L3c of the macaque brain. The raw images for all panels are derived from the Allen Institute website (www.brain-map.org). Bars: 400 µm in (**A**) (for panels **A**–**F**); 400 µm in (**G**) (for panels **G**–**L**); 295 µm in (**M**) (for panels **M**–**R**); 300 µm in (**S**) (for panels **S**–**X**).

**Figure 5 brainsci-14-00372-f005:**
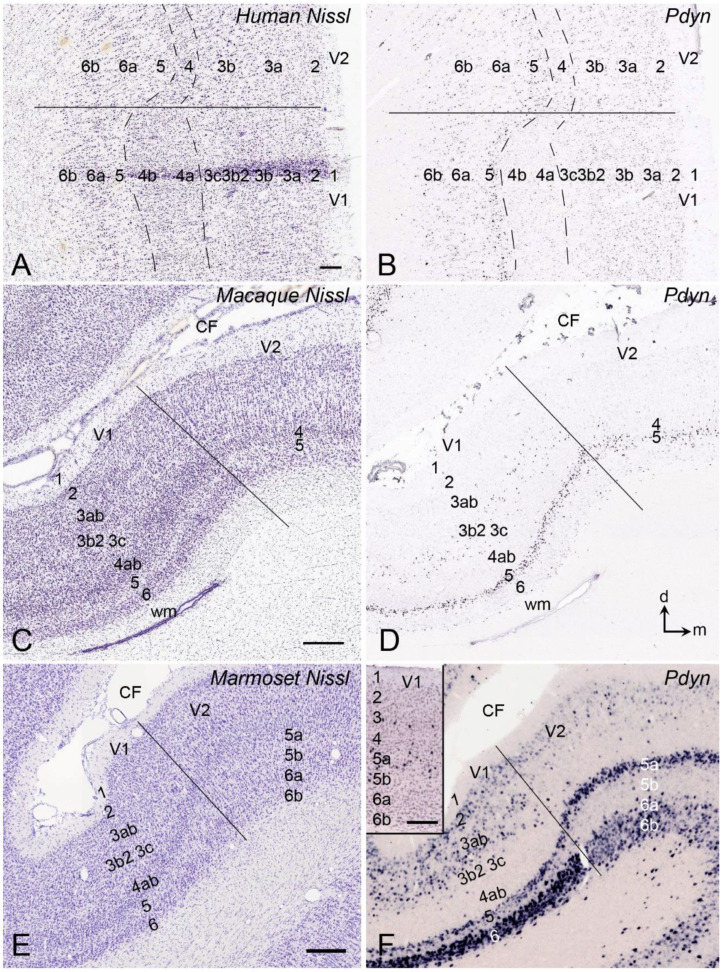
Comparison of *Pdyn* expression in V1 and V2 of the adult human, macaque, marmoset, and mouse brains. (**A**,**B**) Cortical layers and *Pdyn* expression of the human V1 and V2. (**C**,**D**) Cortical layers and *Pdyn* expression of the macaque V1 and V2. (**E**,**F**) Cortical layers and *Pdyn* expression of the marmoset V1 and V2. (Inset in (**F**)) *Pdyn* expression in the mouse V1. Dorsal (d) and medial (m) orientations are indicated in panel (**D**). The raw images for all panels are derived from the Allen Institute website (www.brain-map.org). Bars: 200 µm in (**A**) (for (**A**,**B**); 400 µm in (**C**) (for (**C**,**D**); 210 µm in (**E**) (for (**E**,**F**).

**Figure 6 brainsci-14-00372-f006:**
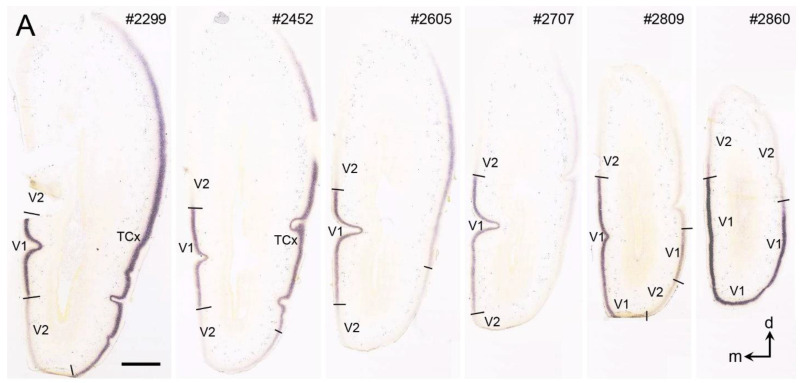
*Npy* expression patterns in V1 and V2 of the prenatal human brain at PCW 21. (**A**) Sequential coronal sections showing the borders and extent of V1 revealed with *Npy* expression patterns. The numbers on the top of each section indicate the section numbers along the anterior–posterior axis of the hemisphere. Dorsal (d) and medial (m) orientations are indicated in section #2860. (**B**,**C**) Two adjacent sections stained for *Npy* and Nissl substance show that the *Npy*-stained band is in the deep portion of the outer cortical plate (CPo). PCW—postconceptional week; MZ—marginal zone; CPi—inner cortical plate; SP—subplate. The raw images for all panels are derived from the Allen Institute website (www.brain-map.org). Bars: 3190 µm in (**A**) (for all panels in (**A**)); 100 µm in (**B**) (for (**B**,**C**)).

**Figure 7 brainsci-14-00372-f007:**
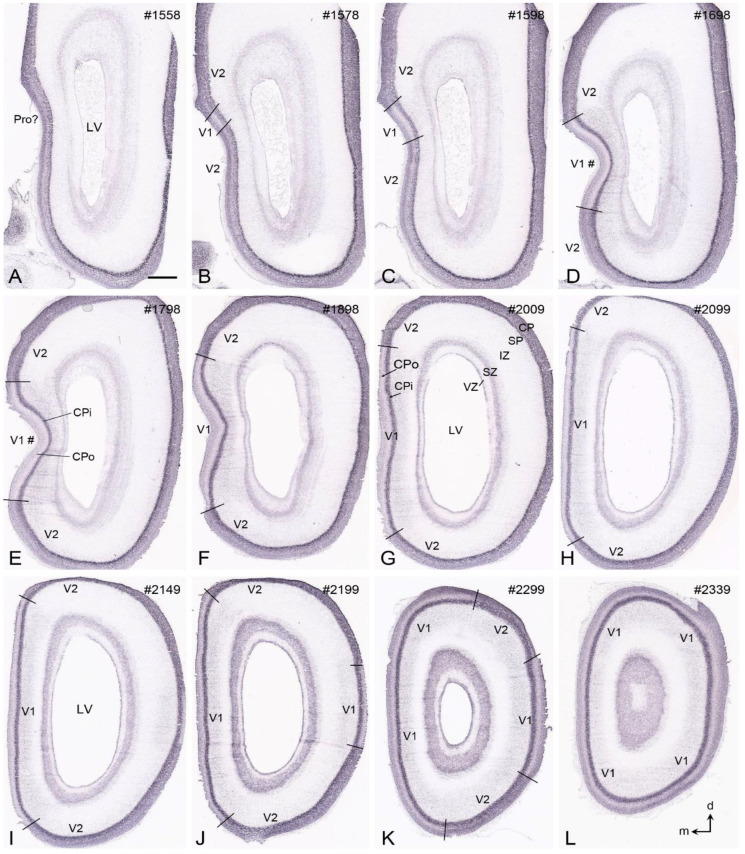
*Enc1* expression patterns in V1 and V2 of the prenatal macaque brain at E90. (**A**–**L**) Sequential coronal sections showing the borders and extent of V1 revealed with *Enc1* expression patterns. The numbers on the top of each section indicate the section numbers along the anterior–posterior axis of the hemisphere. Prenatal cortical layers and dorsal (d)-medial (m) orientations are indicated in panels (**G**) and (**L**), respectively. Compared to the adjoining V2, V1 displays a much weaker *Enc1* expression in the outer cortical plate (CPo in panels (**E**,**G**)), making V1 stand out. E—embryonic day; LV—lateral ventricle. The raw images for all panels are derived from the Allen Institute website (www.brain-map.org). Bar: 1580 µm in (**A**) (for all panels).

**Figure 8 brainsci-14-00372-f008:**
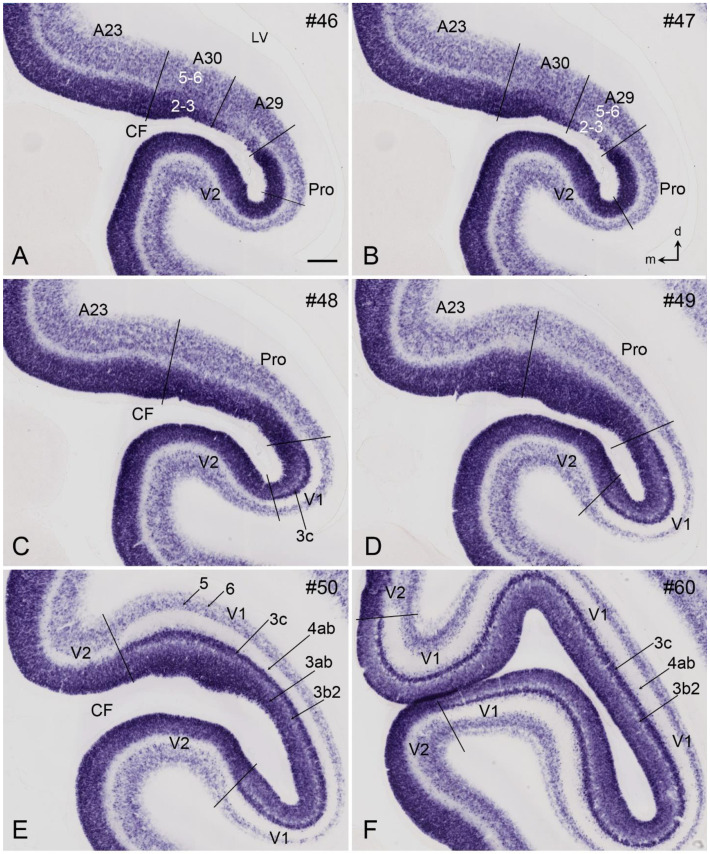
*Enc1* expression patterns in V1 and V2 of the newborn marmoset brain. (**A**–**F**) Sequential coronal sections showing the borders and extent of V1, and adjoining regions revealed with *Enc1* expression patterns. The numbers on the top of each section indicate the section numbers along the anterior–posterior axis of that set of sections stained for *Enc1*. Dorsal (d) and medial (m) orientations are indicated in panel (**B**). Differential *Enc1* expression patterns in the retrosplenial areas 29 (A29) and 30 (A30), A23, Pro, V2, and V1 can be appreciated (**A**–**D**). V1 in panels (**C**–**F**) can be easily identified based on the existence of the thick L4ab (negative *Enc1* expression) and the unique L3c (strong *Enc1* expression). Note the overall denser staining in the superficial part of V2 compared to V1. The raw images for all panels are derived from the Marmoset Gene Atlas (http://www.brainminds.jp). Bar: 425 µm in (**A**) (for all panels).

**Figure 9 brainsci-14-00372-f009:**
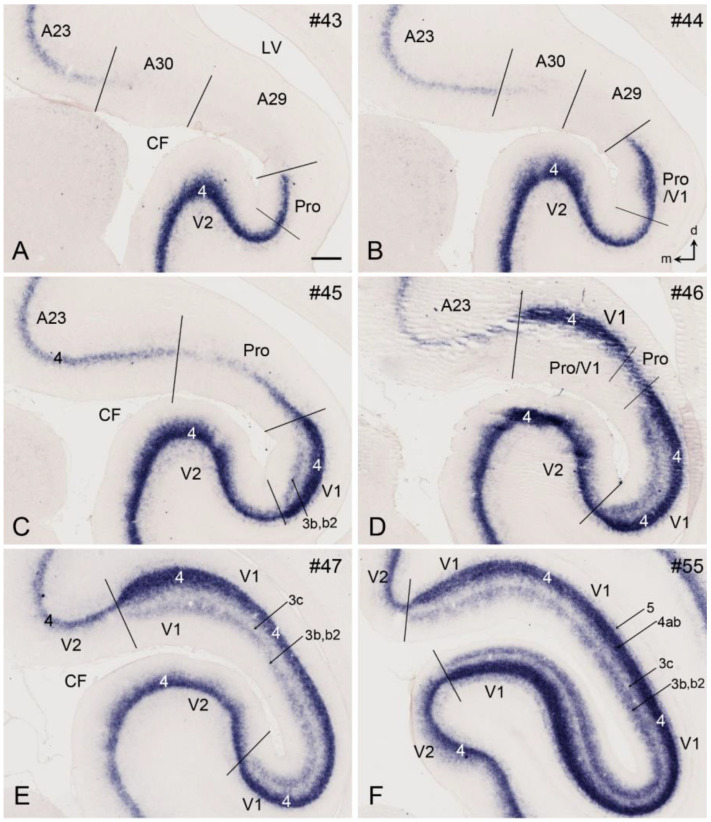
*Rorb* expression patterns in V1 and V2 of the newborn marmoset brain. (**A**–**F**) Sequential coronal sections showing the borders and extent of V1 and adjoining regions revealed with *Rorb* expression patterns. The numbers on the top of each section indicate the section numbers along the anterior–posterior axis of that set of sections stained for *Rorb*. Dorsal (d) and medial (m) orientations are indicated in (**B**). A dense *Rorb*-positive band is seen in L4 of V1, while L3c and L3b-3b2 show faint and moderate *Rorb* expression, respectively. Note the negative labeling in A29 and A30 and weaker labeling in the prostriata (Pro). The raw images for all panels are derived from the Marmoset Gene Atlas (http://www.brainminds.jp). Bar: 425 µm in (**A**) (for all panels).

**Figure 10 brainsci-14-00372-f010:**
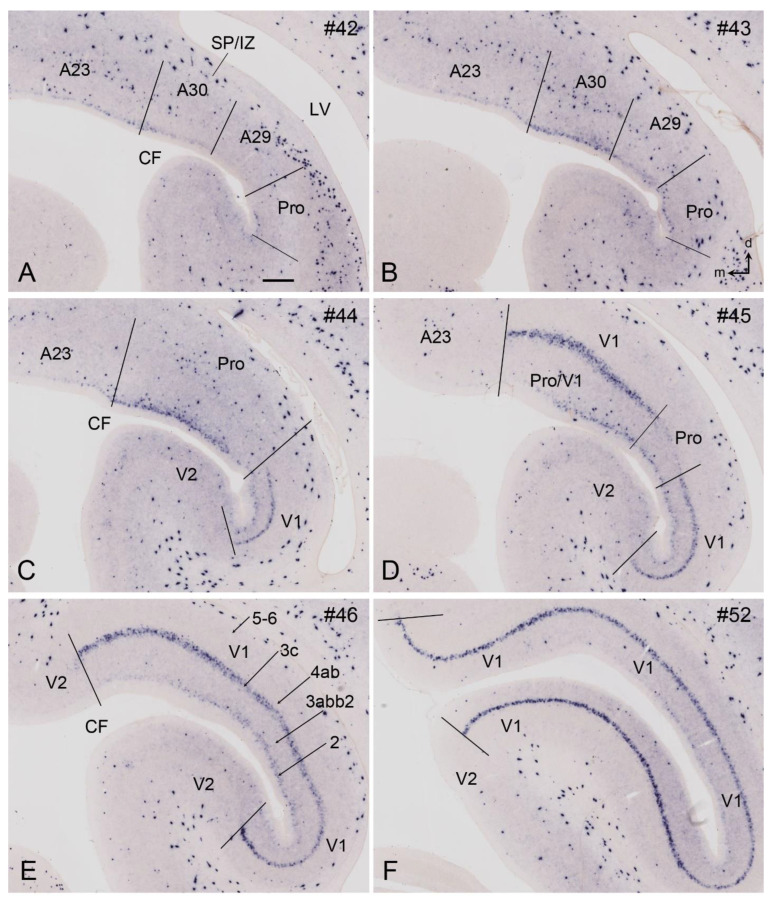
*Npy* expression patterns in V1 and V2 of the newborn marmoset brain. (**A**–**F**) Sequential coronal sections showing the borders and extent of V1 and adjoining regions revealed with *Npy* expression patterns. The numbers on the top of each section indicate the section numbers along the anterior–posterior axis of that set of sections stained for *Npy*. Dorsal (d) and medial (m) orientations are indicated in (**B**). A *Npy*-positive band is clearly seen in L3c of V1. Note that many sparsely distributed but strongly labeled *Npy*-positive cells exist in the subplate and intermediate zone (SP/IZ) of all cortical regions. The raw images for all panels are derived from the Marmoset Gene Atlas (http://www.brainminds.jp). Bar: 425 µm in (**A**) (for all panels).

**Figure 11 brainsci-14-00372-f011:**
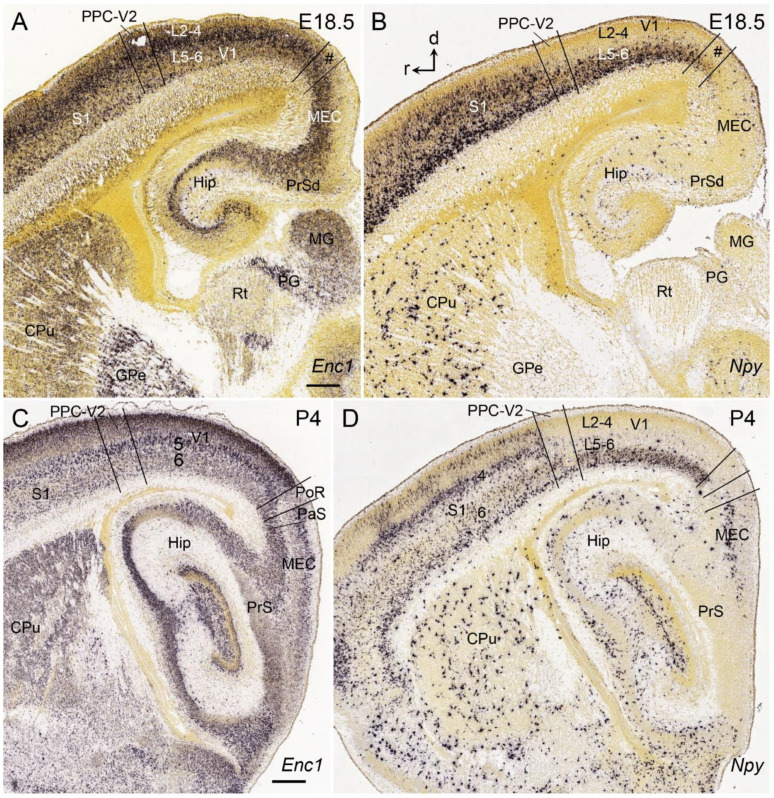
*Enc*1 and *Npy* expression in V1 of the prenatal and postnatal mice. (**A**,**B**) *Enc1* (**A**) and *Npy* (**B**) expression in the sagittal sections of V1 at E18.5. At this age, *Npy* rather than *Enc1* expression in V1 is clearly distinguishable from that in adjoining cortices. Note the strong and weaker *Npy* expression in L6 of V1 and PPC-V2, respectively (**B**). In addition, *Npy* expression is weak in the posteriorly adjoining regions such as the postrhinal–parasubicular cortices (PoR–PaS; indicated by #) and medial entorhinal cortex (MEC). (**C**,**D**) *Enc1* (**C**) and *Npy* (**D**) expression in the sagittal sections of V1 at P4. Overall, the gene expression patterns at P4 are similar to those at E18.5. Rostral (r) and dorsal (d) orientations are indicated in (**B**). PPC—posterior parietal cortex; S1—primary somatosensory cortex; CPu—caudate and putamen; GPe—external globus pallidus; MG—medial geniculate nucleus; PG—pregeniculate nucleus; Rt—reticular thalamic nucleus; Hip—hippocampus; PrS—presubiculum; PrSd—dorsal PrS. The raw images for all panels are derived from the Allen Institute website (www.brain-map.org). Bars: 220 µm in A (for (**A**,**B**)); 330 µm in C (for (**C**,**D**)).

**Figure 12 brainsci-14-00372-f012:**
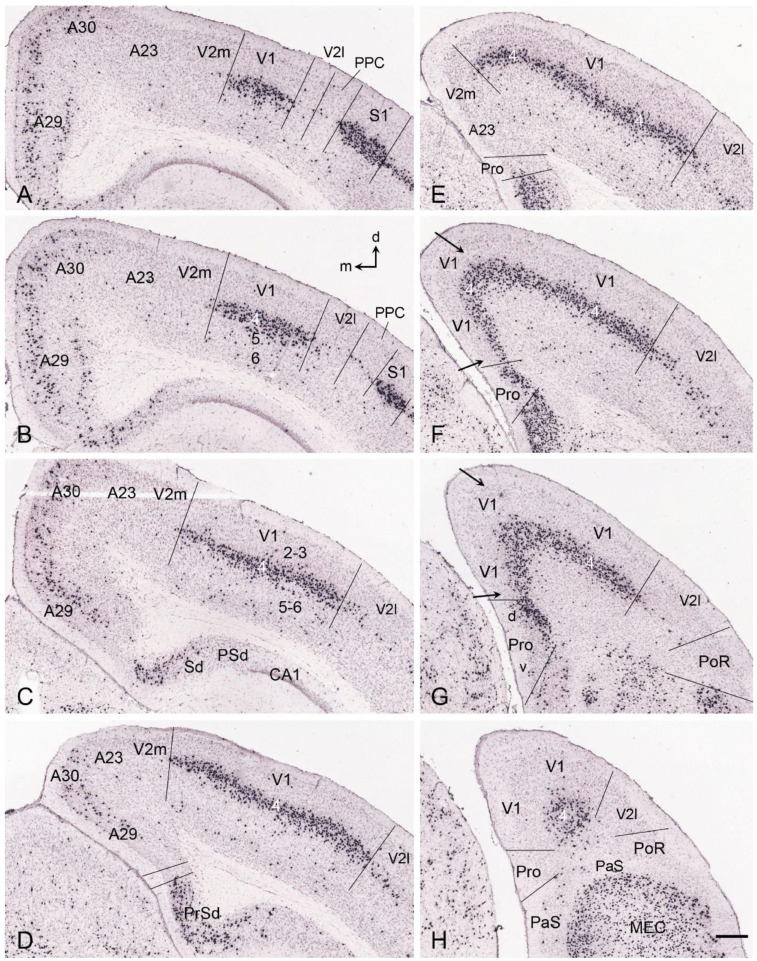
Borders and extent of V1 in coronal sections of the adult mouse. (**A**–**H**) Sequential anterior (**A**) to posterior (**H**) coronal sections showing the borders of V1 with adjoining cortices as revealed with strong *Scnn1a*-*Cre* (*tdTomato*) expression in L4 of V1. Note that at the posterior levels (**F**–**H**), V1 extends medioventrally (indicated by the arrows in panels (**F**,**G**)) and abuts the prostriata (Pro), which has two subdivisions (d and v; see panel (**G**)). Adjoining V2m and V2l contain no or much less *Scnn1a-Cre* expression. Dorsal (d) and medial (m) orientations are indicated in (**B**). The raw images for all panels are derived from the Allen Institute website (www.brain-map.org). Bar: 300 µm in (**H**) (for all panels).

**Figure 13 brainsci-14-00372-f013:**
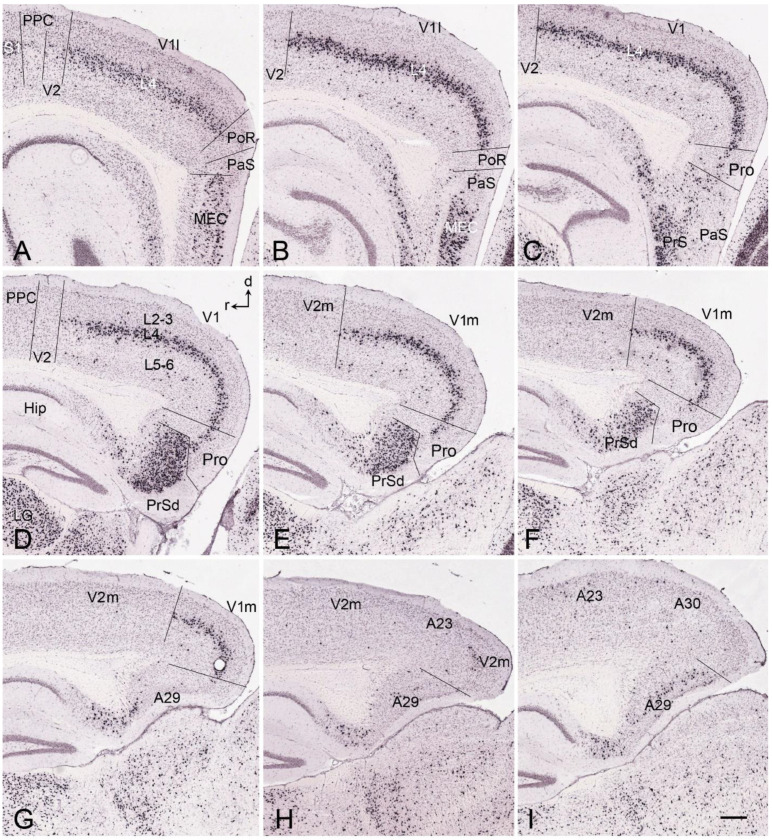
Borders and extent of V1 in sagittal sections of the adult mouse. (**A**–**I**) Sequential middle-lateral (**A**) to medial (**H**) sagittal sections showing the borders of V1 with adjoining cortices as revealed with strong *Scnn1a-Cre* (*tdTomat*o) expression in layer 4 of V1. Note that towards the medial levels (from (**E**) to (**G**)), the posterior V1 gets smaller in size and exists on the medial aspect of the hemisphere, where it adjoins medially with V2m and A23. V2M, A23, and A30 contain no or few *Scnn1a*-*Cre* expressing cells (**G**–**I**). Note that the LG also contains many *Scnn1a-Cre* expressing neurons (**D**). Rostral (r) and dorsal (d) orientations are indicated in (**D**). The raw images for all panels are derived from the Allen Institute website (www.brain-map.org). Bar: 300 µm in (**I**) (for all panels).

**Figure 14 brainsci-14-00372-f014:**
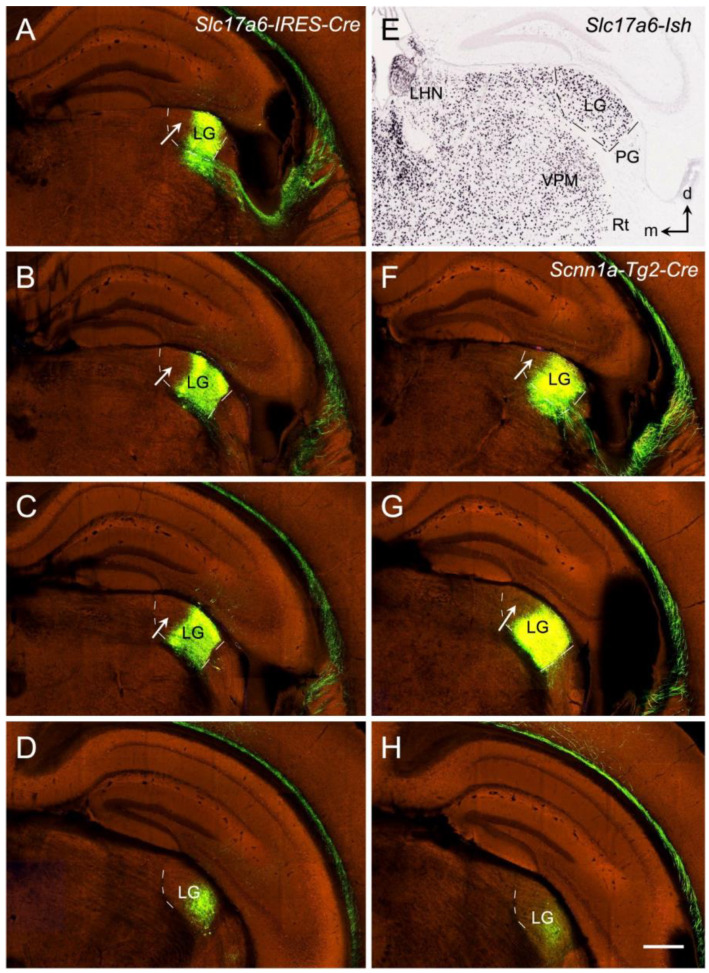
Anterograde viral tracer (rAAV) injection sites in the LG of the Cre-line mice. (**A**–**D**) A tracer injection site restricted in the LG of the *Slc17a6-IRES-Cre* mouse is involved in the ventrolateral part of the anterior and middle LG but not the dorsomedial LG (arrows) and the posterior LG (D and more posterior levels). (**E**) *Slc17a6-Cre* expression in the LG (i.e., DLG) but not in the PG (i.e., VLG) and Rt. (**F**–**H**) A tracer injection site restricted in the LG of the *Scnn1a-Tg2-Cre* mouse is involved in a similar region of the LG as in the *Slc17a6-Cre* mouse. Dorsal (d) and medial (m) orientations are indicated in (**E**). The white arrows point to the LG part without involvement in the injections. VPM—ventroposterior medial nucleus; LHN—lateral habenular nucleus. The raw images for all panels are derived from the Allen Institute website (www.brain-map.org). Bar: 400 µm in (**H**) (for all panels).

**Figure 15 brainsci-14-00372-f015:**
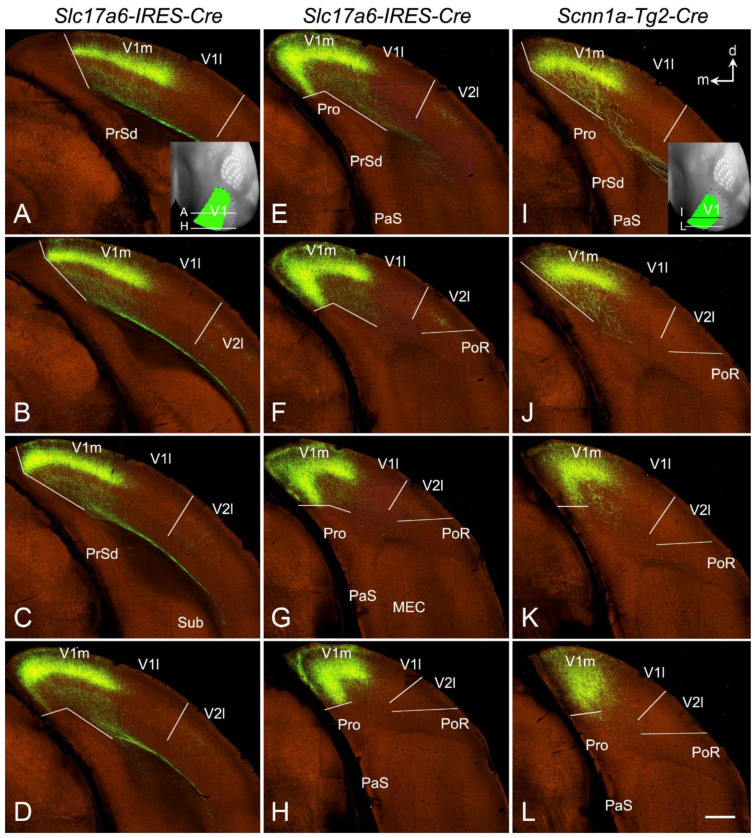
Axon terminal distribution in V1 resulted from the LG injections. (**A**–**H**) Sequential anterior (**A**) to posterior (**H**) coronal sections showing the terminal distribution resulted from the LG injection in the *Slc17a6-Cre* mouse shown in Figure 14A–D. Note that the medial border of the dense terminal band in L4-L3b of V1 corresponds to the border identified with the *Scnn1a-Cre* expression pattern (see Figure 12). The overall distribution of the labeled terminals in V1 is shown on the dorsal aspect of the hemisphere in the inset of panel (**A**). (**I**–**L**) Sequential coronal sections roughly correspond to the sections in panels (**E**–**H**), showing the terminal distribution resulting from the LG injection in the *Scnn1a-Cre* mouse shown in Figure 14F–H. The overall distribution of the labeled terminals in V1 is shown on the dorsal aspect of the hemisphere in the inset of panel (**I**). Note the slight difference in the sectioning angles between the two cases. Dorsal (d) and medial (m) orientations are indicated in panel (**I**). The raw images for all panels are derived from the Allen Institute website (www.brain-map.org). Bar: 400 µm in (**L**) (for all panels).

**Figure 16 brainsci-14-00372-f016:**
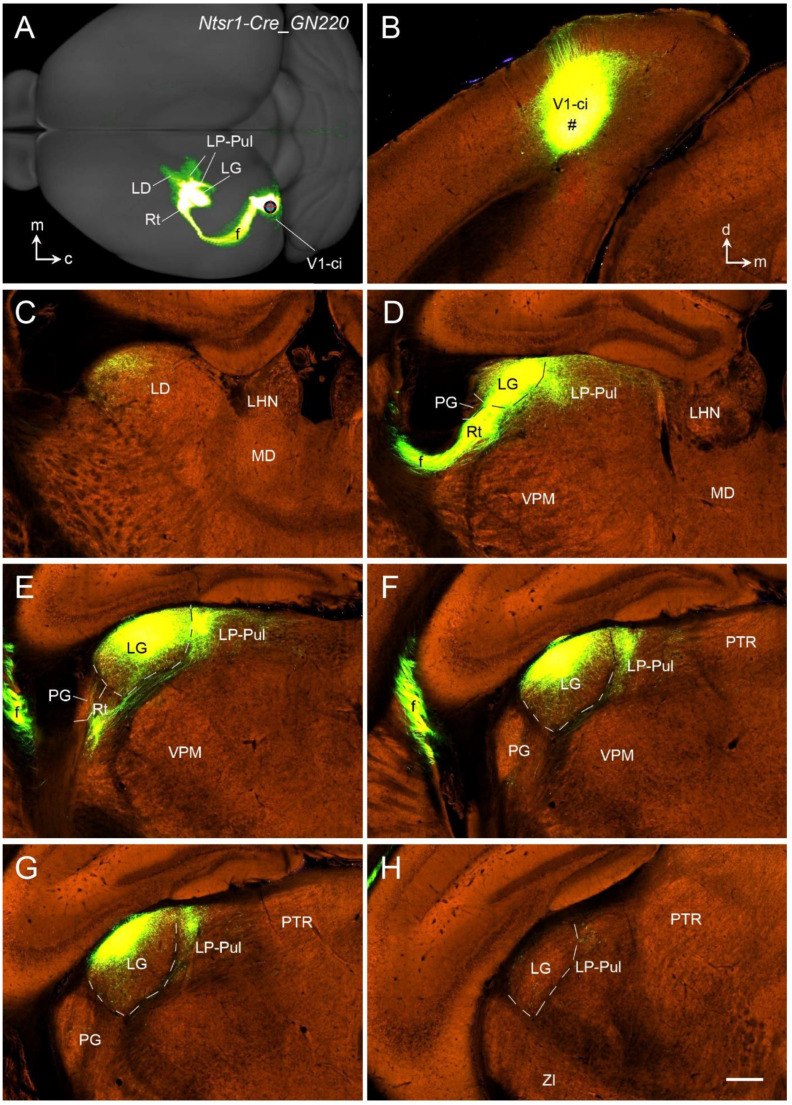
Corticothalamic projections of the caudo-intermediate V1 revealed in an *Ntsr1-Cre_GN220* mouse. (**A**) Dorsal aspect view of the hemisphere showing the injection site (red cross), fiber pathway (f), and terminal fields in the LD, LG, Rt, and LP–Pul. Caudal (c) and medial (m) orientations are indicated. (**B**) An image showing the injection site (#) located in the caudo-intermediate V1 (V1-ci) of the *Ntsr1-Cre* mouse, in which *Ntsr1-Cre* is exclusively expressed in L6 (not in other layers). (**C**–**H**) Sequential anterior (**C**) to posterior (**H**) sections showing labeled axon terminals in the LD, LG, LP–Pul, and Rt. Note that the terminal labeling is mainly seen in the dorsolateral part of the LG and LP–Pul at the anterior levels (**D**–**G**) but not the posterior levels (H and more posterior levels). Dorsal (d) and medial (m) orientations are indicated in panel (**B**) for all histological sections. LD—laterodorsal nucleus; LP–Pul—lateroposterior–pulvinar complex; MD—mediodorsal nucleus; PTR—pretectal region; ZI—zona incerta. The raw images for all panels are derived from the Allen Institute website (www.brain-map.org). Bar: 280 µm in (**H**) (for panels (**B**–**H**)).

**Figure 17 brainsci-14-00372-f017:**
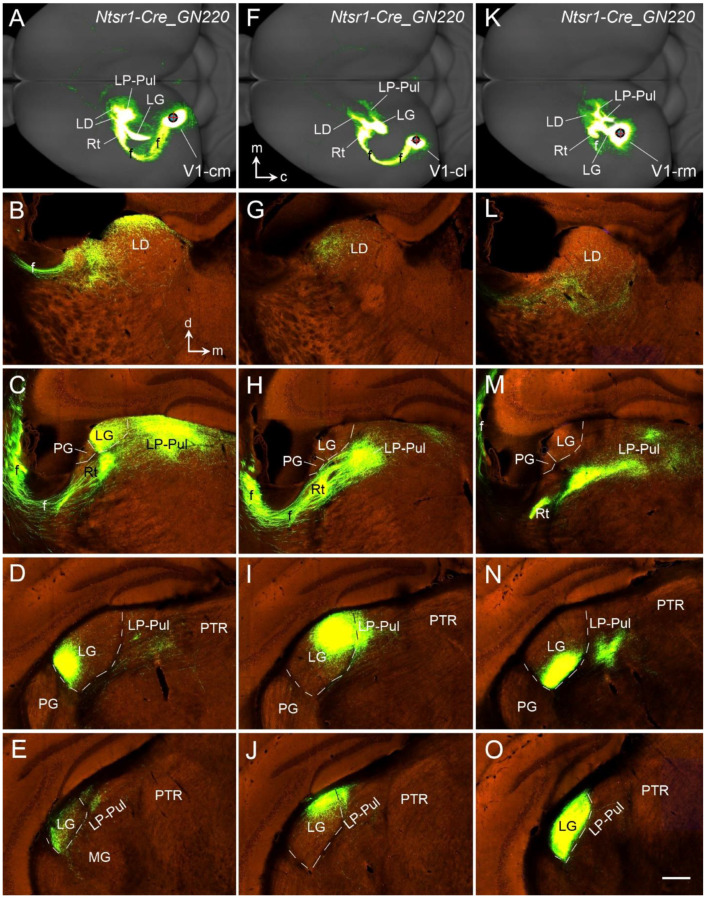
Corticothalamic projections resulted from injections in three other locations in the mouse V1. (**A**–**E**) An injection in the caudomedial V1 (**A**) of an *Ntsr1-Cre_GN220* mouse produces labeled axon terminals in the LD, LG, LP–Pul, and Rt (**B**–**E**). Note that the terminal labeling is mainly seen in the ventrolateral part of the LG and the dorsolateral part of the LP–Pul at the anterior levels (**C**–**E**). (**F**–**J**) An injection in the caudolateral V1 (**F**) of an *Ntsr1-Cre_GN220* mouse produces labeled axon terminals in the LD, LG, LP–Pul, and Rt (**G**–**J**). (**K**–**O**) An injection in the rostromedial V1 (**K**) of an *Ntsr1-Cre_GN220* mouse produces labeled axon terminals in the LD, LG, LP–Pul, and Rt (**L**–**O**). Note that the terminal labeling is mainly seen in the ventromedial part of the LG and LP–Pul at the posterior levels (**N**,**O**). Caudal (c) and medial (m) orientations are indicated in panel (**F**) for the dorsal views in panels (**A**,**F**,**K**), while dorsal (d) and medial (m) orientations are indicated in panel (**B**) for all histological sections. The raw images for all panels are derived from the Allen Institute website (www.brain-map.org). Bar: 280 µm in (**O**) (for all histological sections).

**Figure 18 brainsci-14-00372-f018:**
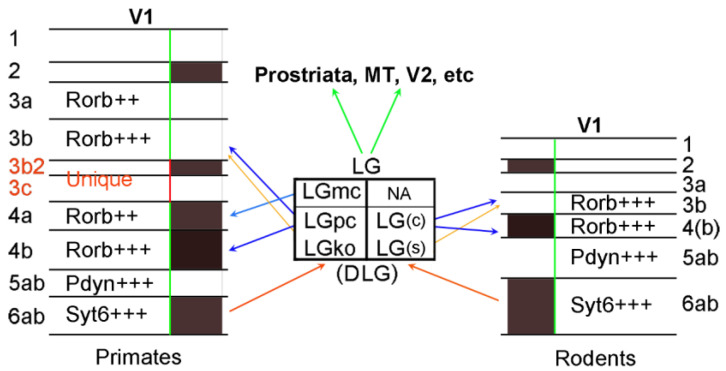
Comparison of the layers and thalamic connections of V1 in primates and rodents. Some conserved molecular markers are indicated for different layers. Cell density in each layer of V1 is coded with black (very dense), dark brown (moderate), and white (less dense or sparse). Rodent LG(DLG) does not appear to have LGmc but has the core and shell [LG(c) and LG(s)], which to some extent appear to correspond to LGpc and LGko of the primates, respectively. Note that if one ignores the unique layers (L3b2 and L3c) in the primates, other layers are overall comparable across species in terms of anatomy, molecular markers, and thalamic connections of V1. In addition, LG (DLG) across species also projects to the extrastriate cortex (prostriata, MT and V2, etc.) but likely with less density. NA—not available. ++ and +++ indicate moderate and strong expression of the genes, respectively.

## Data Availability

The datasets used in this article for human, macaque, and mouse brains are publicly available at the Allen Institute Portal (www.brain-map.org), while those for marmoset brains are accessible at the Marmoset Gene Atlas (http://www.brainminds.jp).

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
