# Peer review of "Lamination, Borders, and Thalamic Projections of the Primary Visual Cortex in Human, Non-Human Primate, and Rodent Brains"

_brainsci, 2024, doi:10.3390/brainsci14040372_

Round 1

Reviewer 1 Report

Comments and Suggestions for Authors

Abstract:

“The primary visual cortex (V1) is one of the mostly studied regions” – of the brain, I suppose?

“…definition of the cortical layers and borders of V1 across high and lower species are very limited” – the use of the terms “higher and lower” with respect to any species is scientifically inappropriate, and reflects an obsolete view of the evolutionary process. You could say “across rodents and primates”

“…consistently link corresponding cortical layers and borders…” – of V1

“Layer 6 of the rodent V1 originates corticothalamic projections to the lateral geniculate, lateral dorsal and reticular thalamic nuclei, and the lateroposterior-pulvinar complex with topographic organization” – isn’t this also true for primates?

“Finally, the direct and strong geniculo-prostriata projections are probably a major contributor to blindsight after V1 lesions.” – does the author actually mean geniculo-extrastriate? Why single out prostriata when a projection from the LGN to this nucleus has not been demonstrated in primates, and it is unknown if rodents show blindsight?

Introduction:

specialization of the middle layers is not obvious or even not exists,” – grammatically incorrect; perhaps “specialization of the middle layers is not obvious and even not exist”

“questioned Brodmann’s L4A and L4B as part of L4 based on comparison of V1 and V2 across NHP species” - questioned whether Brodmann’s L4A and L4B should really be seen as part of L4, based on comparison of V1 and V2 across NHP species. 

Also: historically, the main reason why the assignment of layer 4B as part of layer 4 was questioned, was the fact that cells in this layer form strong extrinsic projections to other areas – something that layer 4 cells don’t do in other areas. Projections to MT were reported by Spatz (PMID: 404175; DOI: 10.1007/BF00239044), Fitzpatrick (PMID: 2418082; DOI: 10.1002/cne.902420408), Palmer (PMID: 16292001; DOI: 10.1093/cercor/bhj078).This needs to be included as part of “The evidence not supporting Brodmann’s L4A and L4B as parts of real L4”

My understanding is that the author agrees that the Hassler scheme of naming V1 layers is more satisfactory than Brodmann’s. The reasons presented appear sound. However, what is the reason for using the designation “3b2”, which (if I recall correctly) was not proposed by Hassler? Wasn’t the corresponding layer named 3B beta?

Section 3 (Laminar organization of V1 in rodents) starts with an attempt to identify homologues of primate V1 and LGN layers in mice, but the arguments are not well developed. Rodent L4 is said to be “more like L4b of the NHPs in terms of the densely packed granular cells”, but the reality is that it does not look like either L4a or L4b in terms of definition (including in Figure 3). The it is said that rodent LG “does not contain an equivalent LGmc”, but no reason or reference for this statement is given. I mean, do we have to choose one? Can’t the rodent L4 be homologous (in terms of developmental origin) to primate L4, but still not be exactly equivalent to either subdivision?

One very obvious difference between mouse and primate layering of V1 is the relative thickness, with layers 2-3 being relatively thin in rodents, while they are very thick (>50% of the volume) in primates. It would be worth considering changes in the relative thickness in this comparative study.

I am not very familiar with the literature on gene expression, so I will not comment on sections 4, 6, 7, 8. I hope another referee can help there. In general, my main comment is that the sections on prenatal animals appear as a completely separate topic, which could be moved to another paper. The topic of layer homologies is complex enough, and adding comprehensive consideration of the developmental changes was (in my view at least) a distraction. This is up to the author, but what I would do is to have two papers, with the second one focused on gene expression during development.

In section 5, it would be useful to consider the differences between the relative sizes of V2 and area prostriata may differ in different primates, with prosimians (Galago) having a relatively large prostriata.

Section 12 may consider the results of a recently published study on the topic or macaque corticothalamic projections.

In section 15, it may be prudent to mention that the evidence of connections from the LGN to prostriata in primates is so far only reliant on tractography, and so it remains pending confirmation with higher resolution methods.

Comments on the Quality of English Language

I have made small suggestions in my main comments but in general the paper is OK with respect to grammar and readability. 

Author Response

Responses to reviewer’s comments

(Please note that revised or newly added contents are highlighted in yellow color.)

Rev. 1

Abstract:

“The primary visual cortex (V1) is one of the mostly studied regions” – of the brain, I suppose?

Reply: Yes. thanks! I have added “of the brain”.

“…definition of the cortical layers and borders of V1 across high and lower species are very limited” – the use of the terms “higher and lower” with respect to any species is scientifically inappropriate, and reflects an obsolete view of the evolutionary process. You could say “across rodents and primates”

Reply:  Thanks for the suggestions! I have used “across rodents and primates”.

“…consistently link corresponding cortical layers and borders…” – of V1

Reply: Yes. I have added “of V1”.

“Layer 6 of the rodent V1 originates corticothalamic projections to the lateral geniculate, lateral dorsal and reticular thalamic nuclei, and the lateroposterior-pulvinar complex with topographic organization” – isn’t this also true for primates?

Reply: Yes. It is true for both rodents and primates. In this revision, I have changed the sentence to “Layer 6 of the rodent and primate V1 originates …”.

“Finally, the direct and strong geniculo-prostriata projections are probably a major contributor to blindsight after V1 lesions.” – does the author actually mean geniculo-extrastriate? Why single out prostriata when a projection from the LGN to this nucleus has not been demonstrated in primates, and it is unknown if rodents show blindsight?

Reply: Thanks for pointing it out! I meant geniculo-extrastriate projections but also emphasized the strong geniculo-prostriata projections, which was also reported in human brains (see section 15). I agree it is unknown if rodents show blindsight (see discussion in section 15). Here in Abstract section, I have revised the sentence as the following.

“Finally, the direct geniculo- extrastriate (particularly the strong geniculo-prostriata) projections are probably major contributors to blindsight after V1 lesions.”

Introduction:

“specialization of the middle layers is not obvious or even not exists,” – grammatically incorrect; perhaps “specialization of the middle layers is not obvious and even not exist”

Reply: Thanks! I have revised the sentence as “specialization of the middle layers is not obvious, or these layers even not exists,”.

“questioned Brodmann’s L4A and L4B as part of L4 based on comparison of V1 and V2 across NHP species” - questioned whether Brodmann’s L4A and L4B should really be seen as part of L4, based on comparison of V1 and V2 across NHP species. 

Reply: Thanks for pointing this out! I have adopted your suggestion.

Also: historically, the main reason why the assignment of layer 4B as part of layer 4 was questioned, was the fact that cells in this layer form strong extrinsic projections to other areas – something that layer 4 cells don’t do in other areas. Projections to MT were reported by Spatz (PMID: 404175; DOI: 10.1007/BF00239044), Fitzpatrick (PMID: 2418082; DOI: 10.1002/cne.902420408), Palmer (PMID: 16292001; DOI: 10.1093/cercor/bhj078). This needs to be included as part of “The evidence not supporting Brodmann’s L4A and L4B as parts of real L4”

Reply: Thanks for mentioning this. This reason has been added in the text. The three papers you mentioned have also been included in this revision.

My understanding is that the author agrees that the Hassler scheme of naming V1 layers is more satisfactory than Brodmann’s. The reasons presented appear sound. However, what is the reason for using the designation “3b2”, which (if I recall correctly) was not proposed by Hassler? Wasn’t the corresponding layer named 3B beta?

 Reply: Thanks for mentioning this minor change. The reason for using “3b2” rather than “3B beta” is simply for simplifying the layer term. In this revision, I have indicated that “layer 3b2 is is a simplified term for layer 3B beta of Hassler’s term”.

Section 3 (Laminar organization of V1 in rodents) starts with an attempt to identify homologues of primate V1 and LGN layers in mice, but the arguments are not well developed. Rodent L4 is said to be “more like L4b of the NHPs in terms of the densely packed granular cells”, but the reality is that it does not look like either L4a or L4b in terms of definition (including in Figure 3). The it is said that rodent LG “does not contain an equivalent LGmc”, but no reason or reference for this statement is given. I mean, do we have to choose one? Can’t the rodent L4 be homologous (in terms of developmental origin) to primate L4, but still not be exactly equivalent to either subdivision?

Reply:  Thanks for the questions! My understanding is that both L4b of the primates and L4 of the rodents contain densely packed granular (small) cells (see Nissl stains in Fig. 3 for rodents and Fig. 4 for primates). This is one of the definition criteria for L4 and one of the reasons I try to link the L4b in primates to L4 in rodents (there are also other reasons, see related text). As for “rodent LG does not contain an equivalent LGmc”, it is clearly shown in rodent literature and brain atlases [cited Wang et al., 2020), and connectional data also support this (see related text). Anyway, it is a minor issue although I am not 100% sure. Alternatively, it is also ok if we just treat the rodent L4 as homologous to primate L4, and this is why I have labelled whole rodent L4 as L4 rather than L4b.

One very obvious difference between mouse and primate layering of V1 is the relative thickness, with layers 2-3 being relatively thin in rodents, while they are very thick (>50% of the volume) in primates. It would be worth considering changes in the relative thickness in this comparative study.

Reply: Yes. The changes in the relative thickness of the layers have been mentioned in the text (at the end of section 3).

I am not very familiar with the literature on gene expression, so I will not comment on sections 4, 6, 7, 8. I hope another referee can help there. In general, my main comment is that the sections on prenatal animals appear as a completely separate topic, which could be moved to another paper. The topic of layer homologies is complex enough, and adding comprehensive consideration of the developmental changes was (in my view at least) a distraction. This is up to the author, but what I would do is to have two papers, with the second one focused on gene expression during development.

Reply: Thanks for the comments! Since this paper also discuss the V1 borders across species (as the title suggests), I believe it is also proper in this paper to discuss the V1 borders in developing animals.

In section 5, it would be useful to consider the differences between the relative sizes of V2 and area prostriata may differ in different primates, with prosimians (Galago) having a relatively large prostriata.

Reply: thanks for the suggestion! I would love to do this. However, it appears that no systematic studies have been carried out to reliably measure the relative sizes of V2 and area prostriata in different primates. In addition, this study mainly focuses on V1 (as the title suggests) rather than V2 and area prostriata.

Section 12 may consider the results of a recently published study on the topic or macaque corticothalamic projections.

Reply: Thanks! I have tried my best to include as many literatures as I can regarding macaque V1 corticothalamic projections. However, I did not include the literatures on corticothalamic projections beyond V1 since this paper focuses on V1.

In section 15, it may be prudent to mention that the evidence of connections from the LGN to prostriata in primates is so far only reliant on tractography, and so it remains pending confirmation with higher resolution methods.

Reply: thanks! I have mentioned that in this revision.

Comments on the Quality of English Language

I have made small suggestions in my main comments but in general the paper is OK with respect to grammar and readability. 

Reply: Thanks for that!

Reviewer 2 Report

Comments and Suggestions for Authors

Manuscript “Lamination, borders and thalamic projections of the primary visual cortex in human, non human primate and rodent brains” devoted to the peculiarities of the primary visual cortex laminar organization and areas V1-V1 border definition in primates and rodents. A developmental aspect of these definitions was also reviewed. This issue is important for researchers in broad areas on neuroscience, including neuroanatomy and neurophysiology. The main text is well structured and highly informative.

Major: Why primary visual cortex of primates and rodents (nor carnivores) were compared? This point should be added to the Introduction.

Minor:

Introduction:

1.    You should present histological features of the Brodmann’s cortical layers definition at the beginning of the review.

2.    You should give clear differences between the meaning of capital letter and lowercase letter in the layers naming (“L4A, L4B,” or “L4b and L4b”, etc).

3.    “L4A and L4B do not receive strong inputs from the LG DLG, as the real 4ab does”. Please, disclose an abbreviation.

4.    “Finally, it should be mentioned that L4A and L4B appear to be additional layers unique to primate V1 and thus they also do not belong to typical L3”. L3 or L4?

5.    “Furthermore, since the mouse LG [usually termed dorsal LG (DLG) in rodent literature…”. In carnivore’s and primate’s dorsal lateral geniculate nucleus also can be abbreviated as DLG (or DLGN). Please clarify it.

6.    What the reason to use single data about newborn marmoset? No explanation exists in the main text.

7.    Figure 13 is blurred.

Author Response

Responses to reviewer’s comments

(Please note that revised or newly added contents are highlighted in yellow color.)

Rev 2:

Manuscript “Lamination, borders and thalamic projections of the primary visual cortex in human, non human primate and rodent brains” devoted to the peculiarities of the primary visual cortex laminar organization and areas V1-V1 border definition in primates and rodents. A developmental aspect of these definitions was also reviewed. This issue is important for researchers in broad areas on neuroscience, including neuroanatomy and neurophysiology. The main text is well structured and highly informative.

Reply: Thanks for the nice comments!

Major: Why primary visual cortex of primates and rodents (nor carnivores) were compared? This point should be added to the Introduction.

Reply: Thanks! I have mentioned the reason in the Introduction section (cited below).

“Since several recent BRAIN Initiative Cell Atlas Network (BICAN) projects (funded by NIH) focus on the comparison of transcriptomic cell types in human, NHP and rodent brains, this review article, as a starting point of harmonizing brain ontology across these species, focuses on unifying the definition of cortical layers and borders of V1 in these species. Accordingly, other species such as carnivores are included in this article.”

Minor:

Introduction:

  1. You should present histological features of the Brodmann’s cortical layers definition at the beginning of the review.

Reply: Thanks for the suggestion! In this revision, I have described histological features of Brodmann’s cortical layers in V1 at the beginning of section 2 (cited below).

“According to Brodmann, V1 in humans and NHPs possesses a thick and laminated layer 4 (L4; termed inner granular cell layer), which contains many small and densely packed cells in its outer (L4A) and inner (L4C) parts separated by a sparsely packed intermediate part (L4B), which contains large cells. L5 is located immediately below this L4. L5 (termed ganglion cell layer) is thin and occupied by many large ganglion (pyramidal) cells with low packing density. Below L5 is the L6 (termed multiform cell layer) containing cells with a variety of morphology. L6 consists of an outer part with high packing density of triangular cells (L6A) and an inner part with low packing density of multiform cells (L6B). L3 (termed pyramidal cell layer) is located immediately above L4. L3 is thick and contains many larger pyramidal neurons. L2 (termed outer granular cell layer) is thin and occupied by small pyramidal cells located between the cell-less L1 (termed molecular layer) and the thick L3 [13].”

  1. You should give clear differences between the meaning of capital letter and lowercase letter in the layers naming (“L4A, L4B,” or “L4b and L4b”, etc).

Reply: Thanks! I have done it as suggested (cited below).

 “Note that the capital letters A, B and C indicate the sublayers of V1 in Brodmann’s and Hassler’s terminology while the lower-case letters a, b and c indicate the sublayers of V1 in the present study.””

  1. “L4A and L4B do not receive strong inputs from the LG DLG, as the real 4ab does”. Please, disclose an abbreviation.

Reply: Thanks! I have done it as suggested.

  1. “Finally, it should be mentioned that L4A and L4B appear to be additional layers unique to primate V1 and thus they also do not belong to typical L3”. L3 or L4?

Reply: Thanks! Here it means typical L3 (L3b).

  1. “Furthermore, since the mouse LG [usually termed dorsal LG (DLG) in rodent literature…”. In carnivore’s and primate’s dorsal lateral geniculate nucleus also can be abbreviated as DLG (or DLGN). Please clarify it.

Reply: Thanks. I have mentioned that “sometimes the primate’s LG is also termed DLG” (see section 3).

  1. What the reason to use single data about newborn marmoset? No explanation exists in the main text.

Reply: Thanks for mentioning this. It is because no data is available for prenatal marmoset V1 borders. I have mentioned this at the beginning of section 8.

  1. Figure 13 is blurred.

Reply: Sorry! It is a little blurred in Words file, but in original format (tiff file), the quality of this figure is better. I can send the original tiff file when needed.

Reviewer 3 Report

Comments and Suggestions for Authors

The paper “Lamination, borders and thalamic projections of the primary visual cortex in human, non human primate and rodent brains” aimed to elucidate the laminar properties of the visual cortex in different species of mammals. To this aim, the author reviewed previous papers and analyzed several open-access datasets. The investigation sounds timely and worth, the main outcomes seem novel, and the manuscript looks well written and offers detailed descriptions of the findings. To better locate the study in a generalized framework, there might be room to provide examples of eventual possibilities to implement the results of the present study into concrete applications.

1) Title – Please consider changing “non human” with “non-human”.

2) Abstract – Please consider changing “studies aimed to harmonize” with “studies aiming to harmonize”

3) Abstract – Please consider adding a conclusive sentence about how the properties of the other animals’ V1 relate to those of human V1.

4) Discussion – Overall, the manuscript remains mostly at the descriptive level, in that real interpretations of the results seem rather rare. It is suggested to provide more “explanations” of the results in a relevant framework. For example, at some points the author correctly highlights the importance of age and neural maturation. However, a focused overview of the anatomo-functional relationship between neural maturation of the cortical visual system and visual competences seems somehow missing. Given the obvious relationship between anatomy and function, it might be worth providing insights onto possible links between laminar development described in the present study and age-related progression of healthy/aberrant visual skills. Providing a similar, more mechanistic, interpretation of how the maturation of the visual system at the laminar level could explain the onset of visual competences and beyond, could be beneficial to provide the present study with a stronger background.

5) Discussion – The author insightfully illustrates the relationship between laminar properties and blindsight. This approach is very relevant in the context of providing examples of how the present study can help to advance the current knowledge about vision-based diseases. Nevertheless, blindsight is a very particular condition and it is very rare. Conversely, laminar aberrancies have been associated with conditions (e.g. dyslexia) which are much more common (Adler et al PLoS One 2013) and linked to age-related dynamics (Platt et al (2013). Embryonic disruption of the candidate dyslexia susceptibility gene homolog Kiaa0319-like results in neuronal migration disorders. Neuroscience248, 585-593.). These peculiarities at the laminar level are good candidates to constitute the anatomical counterpart of the functional age-related aberrancies that have been demonstrated in the dyslexic brain (Farah et al 2021 Front in Psychology). On this basis, it could be proposed that the laminar characteristics described in the present study may be considered part of the anatomical foundations of subsequent functional development, as it is suggested by the case of dyslexia.

Author Response

Responses to reviewer’s comments

(Please note that revised or newly added contents are highlighted in yellow color.)

Rev 3:

The paper “Lamination, borders and thalamic projections of the primary visual cortex in human, non human primate and rodent brains” aimed to elucidate the laminar properties of the visual cortex in different species of mammals. To this aim, the author reviewed previous papers and analyzed several open-access datasets. The investigation sounds timely and worth, the main outcomes seem novel, and the manuscript looks well written and offers detailed descriptions of the findings. To better locate the study in a generalized framework, there might be room to provide examples of eventual possibilities to implement the results of the present study into concrete applications. 

1) Title – Please consider changing “non human” with “non-human”.

Reply: Thanks! I changed it. 

2) Abstract – Please consider changing “studies aimed to harmonize” with “studies aiming to harmonize”

 Reply: Thanks! I changed it as suggested. 

3) Abstract – Please consider adding a conclusive sentence about how the properties of the other animals’ V1 relate to those of human V1.

Reply: Thanks! I have mentioned that primate and human V1 are similar in terms of the unique middle layers while other layers appear to have similar properties across species.

4) Discussion – Overall, the manuscript remains mostly at the descriptive level, in that real interpretations of the results seem rather rare. It is suggested to provide more “explanations” of the results in a relevant framework. For example, at some points the author correctly highlights the importance of age and neural maturation. However, a focused overview of the anatomo-functional relationship between neural maturation of the cortical visual system and visual competences seems somehow missing. Given the obvious relationship between anatomy and function, it might be worth providing insights onto possible links between laminar development described in the present study and age-related progression of healthy/aberrant visual skills. Providing a similar, more mechanistic, interpretation of how the maturation of the visual system at the laminar level could explain the onset of visual competences and beyond, could be beneficial to provide the present study with a stronger background.

Reply: Thanks for the helpful comments! In this review article, the border-determination and lamination of V1 in both adult and developing animals are mainly focused (as suggested by the title). However, although anatomic-functional relationship between neural maturation of the cortical visual system and visual competences is not a focus of this article, I agree it may be useful to briefly review possible links between laminar development and age-related events in healthy/aberrant visual system. In this revision, I added a paragraph (section 16) to briefly discuss this issue. (cited below).

        “As shown in this article, the specialized L4 is detectable from PCW 21 onward in human V1 (Figure 6), from E70 onward in macaque V1 (Figure 7) and at around birth in marmosets (Figures 8-10) and mice (Figure 11) based on specific gene expression. In human brains, adult-like lamination of V1 is identifiable in Nissl-stained sections from prenatal weeks 29-30 onward [41, 42]. In general, anatomic changes continue throughout postnatal developmental period across species. In monkey visual cortex, for example, the laminar distribution of feedback connections changes in the first two months of life [111, 112]. Similarly, vision-related functions also mature throughout postnatal developmental stages with basic receptive field (RF) properties and visual functions mature earlier than complex ones [112]. In the human visual cortex, the fundamental RF architecture becomes adult-like by age 5 and visuo-spatial functions continue to develop afterward. This finding suggests that, despite the early maturation of the RF structure, functional interactions within and across RFs may mature slowly [113].”

5) Discussion – The author insightfully illustrates the relationship between laminar properties and blindsight. This approach is very relevant in the context of providing examples of how the present study can help to advance the current knowledge about vision-based diseases. Nevertheless, blindsight is a very particular condition and it is very rare. Conversely, laminar aberrancies have been associated with conditions (e.g. dyslexia) which are much more common (Adler et al PLoS One 2013) and linked to age-related dynamics (Platt et al (2013). Embryonic disruption of the candidate dyslexia susceptibility gene homolog Kiaa0319-like results in neuronal migration disorders. Neuroscience248, 585-593.). These peculiarities at the laminar level are good candidates to constitute the anatomical counterpart of the functional age-related aberrancies that have been demonstrated in the dyslexic brain (Farah et al 2021 Front in Psychology). On this basis, it could be proposed that the laminar characteristics described in the present study may be considered part of the anatomical foundations of subsequent functional development, as it is suggested by the case of dyslexia.

Reply: As suggested, two paragraph have been added (see section 16; page 13) to discuss the possible link between the laminar organization and its possible changes in dyslexia. (Cited below). 

    “Laminar aberrancies in development have been associated with an animal model for autism [114] and with disruption of genes associated with developmental dyslexia [115], both autism and dyslexia display obvious dysfunctions in visual skills. For example, embryonic disruption of the candidate dyslexia susceptibility gene homolog Kiaa0319-like results in neuronal migration disorders [116]. These peculiarities at the laminar level are good candidates to constitute the anatomical counterpart of the functional age-related aberrancies that have been demonstrated in the dyslexic brain [117]. On this basis, it could be proposed that the laminar characteristics described in the present study may be considered part of the anatomical foundations of subsequent functional development, as it is suggested by the case of dyslexia.

                   In humans and NHPs, the magnocellular (M) pathway is the major stream of inputs from the retina to LGmc, to V1 and then to the dorsal extrastriate and parietal regions. This pathway mediates the ability to rapidly identify letters and their order because they control visual guidance of attention and of eye fixations. Abnormal development of this pathway could cause dyslexia. Evidence for M cell impairment has been reported at all levels of the visual system [118, 119]. In addition, Treatments that facilitate M function, such as viewing text through yellow or blue filters, can greatly increase reading progress in children with visual reading problems [119]. Since cell loss in LGmc has been reported in patients with dyslexia [120], and LGmc mainly projects to L4a (see section 11), it would be interesting to explore in future whether the thickness of L4 decreases as the disorder progresses.”

Round 2

Reviewer 1 Report

Comments and Suggestions for Authors

The paper has been appropriately revised. no other comments.

Reviewer 3 Report

Comments and Suggestions for Authors

Fine